# MineTheGap: Automatic Mining of Biases in Text-to-Image Models

## Abstract

Text-to-Image (TTI) models generate images based on text prompts, which often leave certain aspects of the desired image ambiguous. When faced with these ambiguities, TTI models have been shown to exhibit biases in their interpretations. These biases can have societal impacts, *e.g.*, when showing only a certain race for a stated occupation. They can also affect user experience when creating redundancy within a set of generated images instead of spanning diverse possibilities. Here, we introduce MineTheGap– a method for automatically mining prompts that cause a TTI model to generate biased outputs. Our method goes beyond merely detecting bias for a given prompt. Rather, it leverages a genetic algorithm to iteratively refine a pool of prompts, seeking those that expose biases. This optimization process is driven by a novel bias score, which ranks biases according to their severity, as we validate on a dataset with known biases. For a given prompt, this score is obtained by comparing the distribution of generated images to the distribution of LLM-generated texts that constitute variations on the prompt.[1]

## 1 Introduction

Text-to-Image (TTI) models generate realistic images according to a natural text prompt. Recent years have seen significant progress in the perceptual quality of the images they generate as well as in the adherence of those images to the text prompts. However, despite substantial progress in these two aspects, state-of-the-art models often fall short of expectations when examining the semantic diversity of the outputs they generate for each prompt (Fig. 1), resulting in a gap between human-expected and model-generated diversity. This gap can introduce prejudiced perspectives, as TTI models learn from datasets that have been shown to embed sociodemographic biases (Birhane et al., 2021; Garcia et al., 2023), leading to the generation of images that reflect or even amplify these biases (Bianchi et al., 2023; Cho et al., 2023). Recent work has taken a broader view by developing methods for detecting open-set biases, surfacing previously unexplored forms of bias. These include analyzing bias for a specific prompt (Chinchure et al., 2025), a specific concept (Rassin et al., 2024), or a larger set of given captions (D'Incà et al., 2024).

In this work, we build on these efforts by moving beyond the evaluation of a TTI model on a limited set of prompts or concepts. We introduce MineTheGap, a method that automatically mines the vast textual space of valid prompts to find those that cause a TTI model to produce biased outputs. MineTheGap is a genetic algorithm based optimization process, that leverages a Large Language Model (LLM). Initially, the LLM is used to generate a set of random candidate prompts for the TTI model, forming the initial population for optimization. Then, at each iteration, all prompts in the current population are measured for bias using our proposed bias score, and the prompts on which the model is found to be most biased are used to generate the population of the next iteration. Upon termination, the most biased prompts are presented to the user.

Mining biased prompts requires a bias measure that ranks prompts from the entire prompt space, that vary substantially in their semantics. Prior approaches are limited to quantifying bias along specific axes, regardless if these are predefined as is typical in studies of sociodemographic biases, or proposed on the fly using LLMs. While such methods have advanced open-set bias detection, they do not provide a unified way to evaluate the overall bias of a prompt across multiple, potentially interacting axes. To overcome this restriction, we introduce a novel measure that compares the semantic

---

[1]Code and examples is provided in an anonymous repository

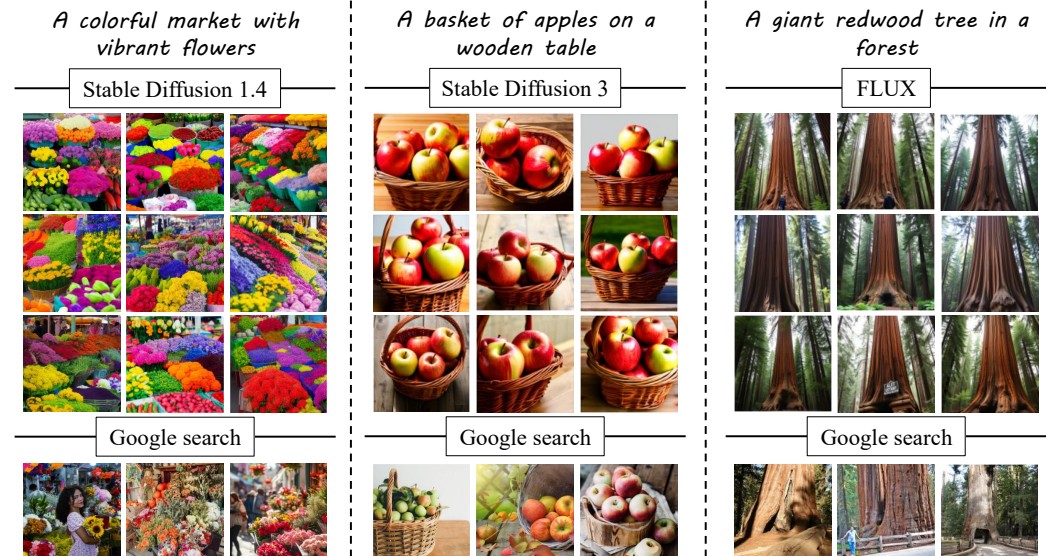

Figure 1: **Limited semantic diversity**. The top panel shows 9 images generated by the specified TTI model for each given prompt. These prompts, identified by MINETHEGAP, introduce bias in the generated images. In contrast, the bottom panel presents 3 images retrieved through a Google Photos search, illustrating alternative interpretations of the same prompt.

distribution of generated images to an intended distribution of plausible interpretations. Since this intended distribution is fundamentally unobservable, we approximate it by sampling diverse textual variations from an LLM, shifting the focus from fixed axes to a range of semantic interpretations.

Consider, for example, the prompt "A photo of a nurse", for which 20 images generated by three TTI models are shown in Fig. 2, alongside images generated from textual variations that span additional semantic options. All models exhibit bias by predominantly producing female figures, and further deviations emerge, as the vast majority of images generated by Stable Diffusion (SD) 2.1 (Rombach et al., 2022) appear in grayscale, while FLUX.1 Schnell (forest labs, 2025) frequently depicts nurses wearing masks. Measuring these deviations and applying appropriate normalization, allows us to rank bias severity across prompts, as we validate on a dataset with known biases. This guides our prompt mining process. While our approach operates in open-set scenarios, we also demonstrate how it can be tailored to target specific types of bias. Automatically uncovering biased prompts highlights model deficiencies in provoking biases, and should lead to further understanding of how to train and use the model when aiming for results that are both fair and beneficial for the user.

To summarize, this work makes two key contributions: (*i*) It introduces an automatic method for revealing biased prompts in TTI models, enabling the discovery of both well-known and previously unseen open-set biases. (*ii*) It proposes a new bias measure that moves beyond axis-based approaches, which depend on defined attributes and categories. Instead, our bias measure compares the distribution of generated images to that of plausible textual variations of a prompt, efficiently capturing deviations from the expected distribution.

## 2 RELATED WORK

**Bias in TTI models.** Bias in TTI models has been a growing concern, with various approaches developed to detect and mitigate biased outputs focusing on particular types of well-defined biases such as race and gender (Luccioni et al., 2024; Clemmer et al., 2024; Gandikota et al., 2024). These methods require prior knowledge of the specific biases investigated. To move beyond predefined biases, open-set bias detection methods aim to uncover biases without the need to pre-define them (Kim et al., 2024; Chinchure et al., 2025). Other approaches tackle this from the perspective of output diversity (Orgad et al., 2023; Sadat et al., 2024; Rassin et al., 2024). Regardless of the de-

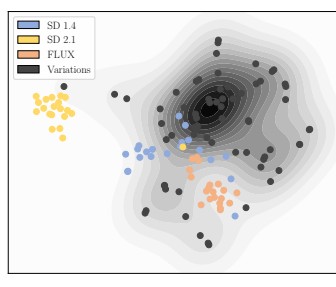 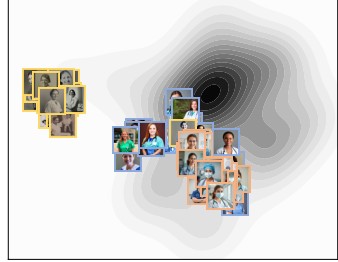 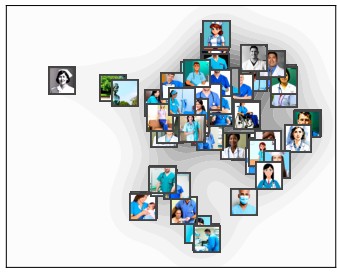

|                      |                      |                                |
|:--------------------:|:--------------------:|:------------------------------:|
| (a) t-SNE visualization | (b) Prompt generations | (c) Prompt variation generations |

Figure 2: **t-SNE visualization of images generated with the prompt "a photo of a nurse" and its variations.** Each point in (a) represents a CLIP embedding of a generated image, projected into 2D space using t-SNE. The gray points, along with their overlaid kernel density estimate, provide a desired reference distribution of images. These images (shown in (c)) were generated using SD 3 with textual variations of the prompt that specify gender, race, style, and surroundings. Inspecting the generations of three TTI models to the prompt (images shown in (b)) reveals that all of them show bias, predominantly generating female figures. However, SD 2.1 tends to further produce grayscale images and FLUX frequently depicts nurses wearing masks, indicating model-specific tendencies.

tection method, recent advancements enable their mitigation (Kim et al., 2024; Parihar et al., 2024). Closest to our work is D'Incà et al. (2024) which introduces a pipeline for evaluating open-set biases in TTI models, applied on a set of real textual captions, such that the evaluated biases depend on this set of captions and are restricted to it. Our method complements these efforts by proposing an automatic method for identifying prompts that cause a TTI model to generate biased outputs, from the unconstrained textual prompt space. This is achieved without relying on predefined captions, concepts, or specific bias categories, yet it can also be adapted to target them when desired.

**Genetic algorithms.** Genetic algorithms are gradient-free optimization processes where a population of candidate solutions iteratively improves by selecting the best candidates to generate the next population. They have been widely applied to prompt optimization tasks (Xu et al., 2022; Tanaka et al., 2023; He et al., 2024), including in the context of TTI models, where they are used to refine prompts for improved image quality and text-image consistency (Pavlichenko & Ustalov, 2023; Tran et al., 2023; Mañas et al., 2024). Broadly speaking, these methods remain within a constrained search space refining a given prompt. Our approach differs by searching the entire prompt space of TTI models to uncover prompts that inherently induce biased outputs of the TTI model.

## 3 AUTOMATIC MINING

MINETHEGAP is a genetic algorithm-based approach that optimizes over the high-dimensional discrete space of valid prompts for a TTI model, denoted by $\mathcal{P}$. Since the space of prompts is non-differentiable, we formulate this as a gradient-free optimization problem, refining a population of candidate prompts through iterative selection and mutation. In this section we introduce our general framework for optimizing over $\mathcal{P}$, which is applicable to any objective. Section 4 discusses our proposed bias objective, used throughout the paper.

Our algorithm employs an evolutionary search procedure that iteratively refines a population of $b$ prompts by selecting those that expose significant bias while introducing new candidates to maintain exploration. The optimization runs for a fixed number of iterations, where at each iteration the best prompts are compared to the best prompts found thus far, such that the final output consists of the top $K$ prompts identified during the entire process. Figure 3 shows the population of prompts across three iterations, illustrating how the search evolves. The framework consists of the following steps.

**Initial population.** The optimization begins with an initial population of $b$ diverse prompts, designed to broadly cover the space of possible biases. Ideally, this population approximates a uniform sample from the space of valid prompts, $\mathcal{P}$. To generate such a set, we instruct an LLM to return $b$ candidate prompts in a single query. These are then parsed and used as the starting population.

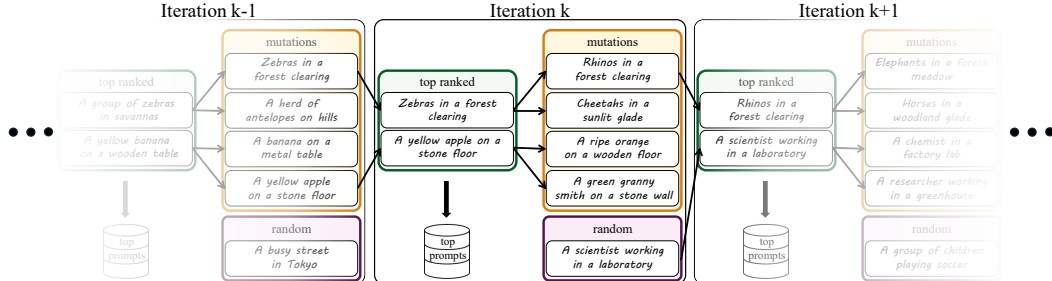

Figure 3: **Mining pipeline.** Three iterations demonstrate the optimization evaluation process. In iteration $k$, we rank the population from the previous iteration (represented by the sentences in the orange and purple frames at iteration $k - 1$) and select the top-ranked (shown in the green frame). We then generate mutations for each of the top-ranked sentences and add random sentences to form the next population. This process repeats in the subsequent iteration $k+1$. Throughout all iterations, sentences are retained in the top prompts bucket if their loss improves on previous top prompts.

**Selection.** To identify the fittest prompts in each iteration, we rank them according to an objective function that quantifies the property of interest. Our framework is independent of the objective used, however for mining biases, we apply the objective defined in Sec. 4.1. This function scores each prompt, where lower values indicate a stronger presence of bias. Since the goal is to mine prompts that best optimize this objective, the $s$ prompts with the lowest scores are selected for further refinement, serving as the foundation for the next generation of prompts. Figure 3 illustrates the selection when $s = 2$ prompts are carried over to the next iteration ("top ranked" panel).

**Mutation.** After selecting the most relevant prompts, a mutation step generates new candidates by exploring diverse modifications to these prompts. Unlike simple rephrasings, these modifications introduce changes that should lead to visually distinct outputs. To achieve this, an LLM is instructed to generate $m$ new modifications for each selected prompt which maintain some connection to the original prompt but allow for creative exploration through substitutions of subjects, omissions, or modifications that significantly alter the expected visual result. For example, given the prompt "a doctor", the LLM may generate related but distinct concepts such as "a nurse" or "a dentist", as illustrated in App. C.2. The mutation pane of each iteration in Fig. 3 shows the newly generated mutations, with arrows pointing from each top-ranked prompt to its corresponding mutations.

**Random candidate injection.** In high-dimensional search spaces such as $\mathcal{P}$, genetic algorithms might converge to suboptimal solutions if the population becomes too homogeneous. To avoid this, we inject additional randomly generated prompts at each iteration. These candidates, generated independently by the LLM, introduce genetic diversity into the population and allow the algorithm to explore more areas of the solution space. Since we use the same population size in each iteration, the number of random candidates to inject, denoted by $r$, is given by $r = b - s \times m$.

As mentioned above, our optimization (*i.e.*, mining) procedure is generic in that it can work with any loss, not necessarily only with bias scores. In App. A we illustrate this optimization method on the simple task of mining prompts that produce red, blue and green images. In this example, the loss being minimized is the mean squared error (MSE) between the generated images and a synthetic image of the target color. In this simple setting, our algorithm typically converges in four iterations.

# 4  RANKING BIAS

To guide the selection process in our mining framework, we require a bias ranking function that assigns a score to each prompt. This score should reflect the extent to which the TTI model exhibits bias when generating images for that prompt. Traditional methods for ranking prompts according to the bias they exhibit, either focus on predefined biases or on LLM generated potential biases, and examine the distribution of images for each class associated with the bias. Here, we propose an efficient, fully automatic approach that does not require specifying the nature of the bias in advance and provides an interpretable ranking of prompts.

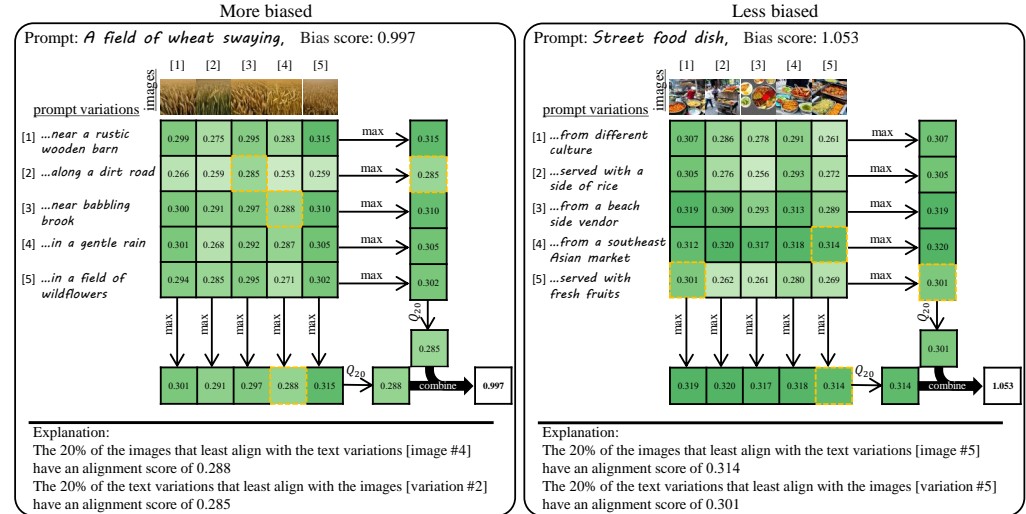

Figure 4: **Measuring bias through least-aligned elements.** Comparison between more (left) and less (right) biased scenarios. In the more biased example, we observe gaps in alignment, where certain textual variations fail to find well-matching images. In contrast, the less biased example demonstrates a more varied distribution of similarity scores, indicating that the TTI model successfully spans the range of plausible interpretations. We set $\alpha = 20^{\text{th}}$ percentile which results in taking the minimum over the maximum similarities.

A key challenge in quantifying bias is determining the expected image distribution for a given prompt. Ideally, we would want to compare with a human-expected distribution, which is unknown. To address this, we leverage an LLM to approximate the expected variations in how a given prompt could be interpreted. Specifically, we use the LLM to generate a set of diverse textual variations of the prompt, designed to span its possible ambiguities. Unlike the mutation step in the mining process, which aims to refine candidate prompts, this process explicitly models the different plausible meanings embedded within a single prompt, providing a reference for evaluating the TTI model's outputs. Examples of textual variations and how they differ from the mutations are given in App. C.

## 4.1 MEASURING BIAS THROUGH LEAST-ALIGNED ELEMENTS

Given a prompt $p$, we generate $N$ images using a TTI model, $\mathcal{I}_p = \{I_1, \cdots, I_N\}$, and $N$ variations of the prompt using an LLM, $\mathcal{V}_p = \{v_1, \cdots, v_N\}$, where $N$ corresponds to the number of images the user wishes to evaluate for each prompt. We then embed both sets into a shared latent space, and compare their distributions. If the set of generated images fails to span the diversity of the textual variations, then this suggests that the model is biased towards certain interpretations of the prompt. To quantify the gap between the two distributions, we construct an $N \times N$ similarity matrix $S$ between the two sets of embeddings, such that element $S_{i,j}$ measures the similarity between the $i^{\text{th}}$ text variation and the $j^{\text{th}}$ generated image, as visualized in Fig. 4. We then construct our bias score (where lower means more biased) as the combination of two complementary scores, as follows.

The *missed visual concepts score* measures how well the generated images cover the diversity of the textual interpretations. For each textual variation, we identify the image most aligned to it by taking the maximum similarity score across all images. Namely, for the $i^{\text{th}}$ text variation, we compute $\max_j S_{i,j}$. A text that is represented in the visual domain has at least one image strongly aligned with it. Now, we want no more than $\alpha$ percent of the text variations to have low maximal alignment scores. We therefore compute the lower $\alpha^{\text{th}}$ percentile of these maximum scores, $Q_\alpha\{\max_j S_{i,j}\}$. The essence of this hyper parameter is to control the sensitivity of the overlap between the two distributions being compared. Setting it to 0 enforces a strict requirement that every textual variation must be represented in the images, while increasing its value gradually relaxes this constraint by allowing a limited number of semantic concepts to differ between the text variations and the images. Similarly, the *least-aligned images score* measures how well each generated image aligns with the

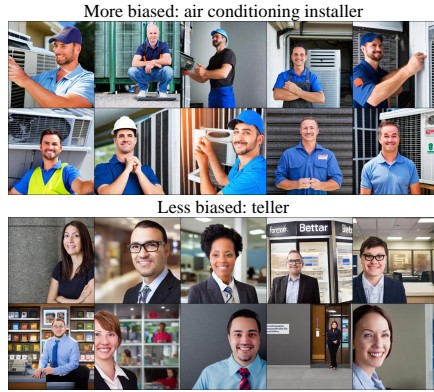

More biased: air conditioning installer

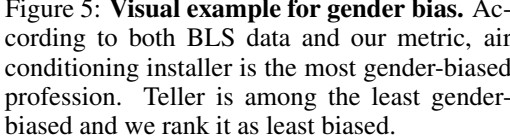

Less biased: teller

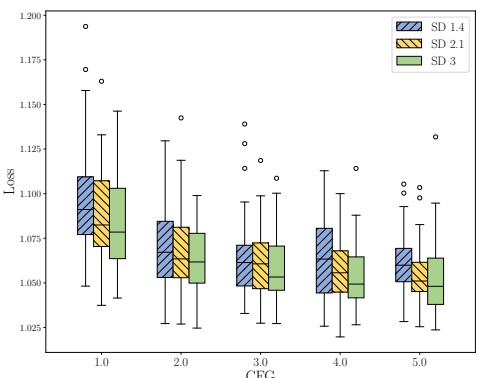

Figure 5: **Visual example for gender bias.** According to both BLS data and our metric, air conditioning installer is the most gender-biased profession. Teller is among the least gender-biased and we rank it as least biased.

Figure 6: **Bias scores across CFG values.** Each box plot summarizes the distribution of measured losses across prompts for a given model and CFG setting. As anticipated, as CFG increases, outputs are scored as more biased.

textual variations. For each image, we compute the maximum similarity across all textual variations, identifying the closest matching text. Namely, for the $j^{\text{th}}$ image, we compute $\max_i S_{i,j}$. Again, we extract the lower $\alpha^{\text{th}}$ percentile among these maxima, $Q_\alpha\{\max_i S_{i,j}\}$, as an indication for the images that have no strong correspondence to any textual interpretation. This identifies cases where the TTI model generates images that do not align with reasonable textual variations, indicating bias.

Finally, to obtain a single relative bias score, we take the average of the missed visual concepts score and the least-aligned images score, and normalize by the mean similarity across the entire matrix,

$$\text{bias} = \frac{\frac{1}{2}\left(Q_\alpha\{\max_j S_{i,j}\} + Q_\alpha\{\max_i S_{i,j}\}\right)}{\frac{1}{N^2}\sum_{i,j} S_{i,j}}. \quad (1)$$

The normalization is needed because the absolute similarity values are meaningful only when comparing semantically close objects. But across different prompts, we are mainly interested in capturing variance within the matrix, rather than absolute magnitudes. This objective provides a single ranking criterion that reflects the severity of bias in the interpretation of a TTI model of a given prompt, and is used in the selection phase of our optimization process.

**Inherent explainability.** The objective described above has an implicit advantage, which is its inherent ability to communicate to a user the source of the bias. This is achieved by identifying the missing visual concepts in the textual variations, and the least-aligned images among the generated outputs. An example is given in the bottom part of Fig. 4 for both high and low bias scores.

## 4.2 VALIDATING OUR BIAS SCORE

To validate our bias score, we assess its ability to capture known biases in TTI models. It is well established that these models often generate stereotypical representations of professions, associating certain occupations with specific genders or ethnicities. A primary source of this phenomenon stems from real-world demographic imbalances that are manifested in the datasets used for training.

Previously, Luccioni et al. (2024) demonstrated that bias in TTI models can be quantitatively ranked and compared to ground-truth occupational statistics, such as those from the U.S. Bureau of Labor Statistics (BLS). Their method focuses on societal biases (*e.g.*, gender and race) and shows a correlation between the ranks assigned by their bias detection framework and the actual gender majority in each profession. Inspired by their evaluation, we apply our bias ranking on the same dataset and evaluate it against the ground-truth majority demographic for each occupation. Specifically, we use a set of images generated by SD 1.4 with the prompt "Photo portrait of X" where X is a profession taken from the BLS (see more details in Luccioni et al. (2024)). Then, we generate 15 variations for each prompt through a Cartesian combination of gender and race attributes. All images and

variations are embedded to a shared latent space where we compare CLIP (Radford et al., 2021), SigLIP (Zhai et al., 2023), and SigLIP-2 (Tschannen et al., 2025). For evaluating the ranking we use Spearman's rank correlation coefficient, $\rho$. Luccioni et al. (2024) achieved $\rho = 0.68$ while our bias score achieves $\rho = 0.72, 0.63, 0.62$ using CLIP, SigLIP and SigLIP 2 respectively. All variants positively correlate with the ground truth, with CLIP showing strongest alignment and exceeding the baseline; we therefore adopt it as the shared latent space. Ten images for professions that our score ranked as highly biased and mostly unbiased are shown in Fig. 5, and App. B.1 presents a comparison between our ranking and the ranking reported by Luccioni et al. (2024).

To further validate our findings, we analyzed how classifier-free guidance (CFG) (Ho & Salimans, 2021) affects the bias score. Increasing CFG is known to reduce sample diversity by favoring image quality and text alignment. For 45 random prompts, we generated sets of 15 images driven by the same random seeds, one set for each CFG value, and computed the bias score against the same set of LLM-generated textual variations. As shown in Fig. 6, across three SD models, higher CFG strength results in more biased outputs, aligning with expectations and reinforcing the validity of our score. A complementary experiment, discussed in App. B.2, demonstrates that applying CADS (Sadat et al., 2024), known to increase sample diversity, yields less biased outputs according to our score.

## 5 EXPERIMENTS

We proceed to evaluate our complete method. MINETHEGAP utilizes an LLM both to drive the optimization process and to create reference texts for bias assessments. For both purposes, we use Llama 3.1-8B-Instruct (Dubey et al., 2024; Meta-AI, 2024) as the core LLM. Comparisons to LLaDA 8B-Instruct (Nie et al., 2025) and Qwen 2.5-7B-Instruct (Yang et al., 2024) are reported in App. D.1, showing the mined prompts are relatively similar while they differ from the captions from COCO. We refer to the prompts used to instruct the LLM on the various tasks as *meta-prompts* and list them in App. C.1. We experiment with four TTI models: SD 1.4, SD 2.1 (Rombach et al., 2022), SD 3 (Esser et al., 2024), and FLUX.1 Schnell (forest labs, 2025), while also showing qualitative results for Qwen-Image (Wu et al., 2025) in App. D.

**Implementation details.** The optimization process runs with a population size of $b = 15$ prompts per iteration, each limited to eight words. At each step, the top $s = 5$ most biased prompts (according to our score) are selected, and each of them undergoes $m = 2$ mutations. To complete the quota, $r = 5$ random prompts are injected in each iteration. An ablation study of these choices is provided in App. A.1. To ensure robustness in our findings, we run 50 independent mining processes per model, each starting with a different randomly sampled initial population using seeds 0 through 49. When calculating the bias score, we generate $N = 15$ images and text variations for each prompt. The least-aligned images score and missed visual concepts score are computed using the $4^{\text{th}}$ smallest similarity value from the ordered set of 15, corresponding to approximately the $25^{\text{th}}$ percentile.

**Mining open-set biases.** Figure 7 shows qualitative results of images generated with mined prompts. For each prompt, we also present a visualization of the text variations that attained the lowest alignment score (*i.e.*, the lower $\alpha^{\text{th}}$ percentile of similarity scores), by showing images generated by the TTI model for those texts. These examples highlight cases in which the model produces biased outputs, repeating semantic options where other visual concepts are applicable. Furthermore, it is evident that when the TTI model is fed with the textual variation as the prompt, it does generate images that depict the additional concepts. This indicates that even though the TTI model is capable of generating images that exhibit a wide range of interpretations to the prompt, it is often biased towards specific interpretations.Appendix D presents additional results, providing a broader perspective on the types of prompts identified as biased. To illustrate the mining process, Fig. 8 shows a segment of an optimization, plotting candidate prompt scores across iterations to highlight how the process refines the prompt pool toward increasingly biased prompts. To assess the effectiveness of the optimization process, we compare the bias scores of prompts mined by multiple runs of MINETHEGAP to those of randomly sampled prompts from the initial population, and to captions from COCO (Lin et al., 2014). As shown in App. A, mined prompts consistently attain lower losses, indicating that they induce stronger bias according to our metric. Furthermore, the variance in bias scores is significantly smaller for mined prompts, suggesting that the genetic search strategy converges to a stable set of highly biased prompts.

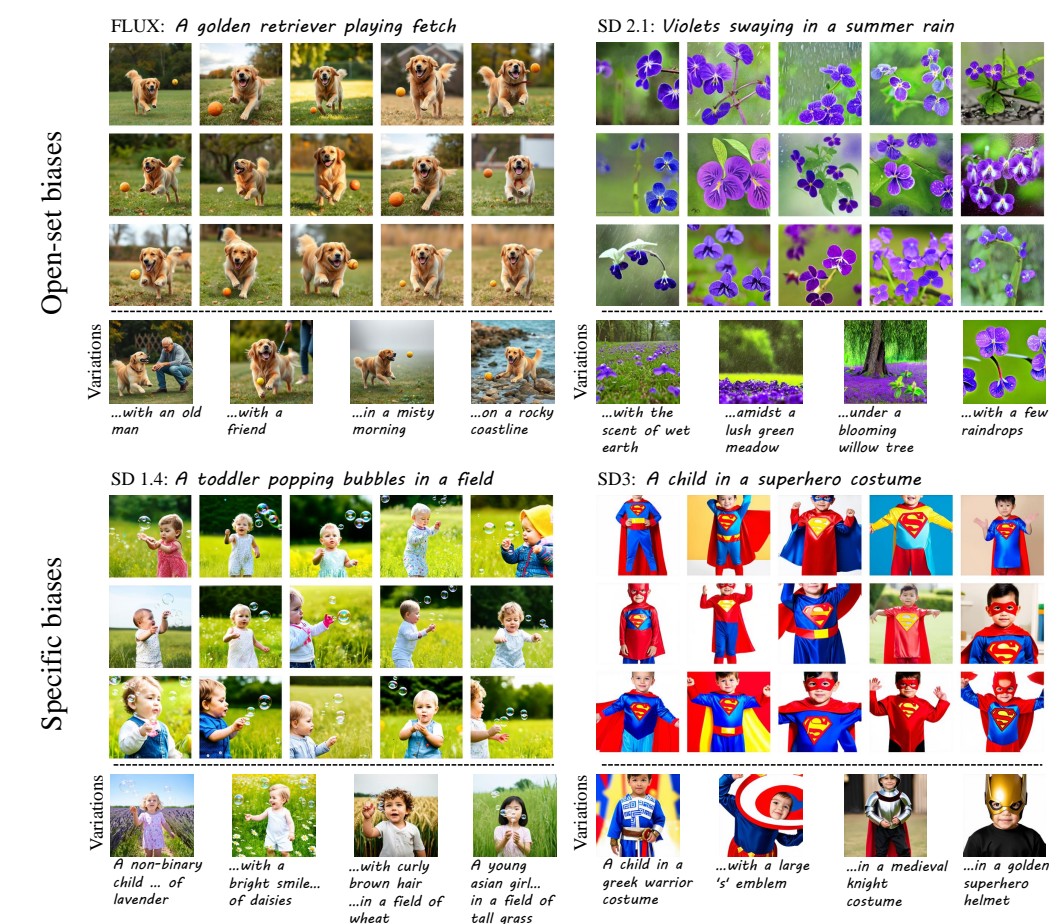

Figure 7: **Mined biased prompts for four TTI models.** For each model, 15 images were generated using a mined prompt. The resulting images are highly similar, exhibiting repetitive semantics. At the bottom of each example, we show images generated from text variations to illustrate the additional concepts the TTI model should incorporate. Top pane illustrates open-set mining showing that in FLUX.1 Schnell, the dog consistently runs toward the camera with no other figures present, and SD 2.1 consistently zooms in on violets. Prompts in bottom pane were mined specifically on gender, race, and age (left) or clothing (right). SD 1.4 generated Caucasian toddlers with blond hair, while associated variations exhibit greater demographic diversity, including children of different races and hair colors. Similarly, SD 3 generates only superhero costumes of Superman.

**Mining specified biases.** Interestingly, we show that by simply adjusting the meta-prompts of the LLM, we can restrict MINETHEGAP to detect biases of a particular type or on specific subjects. More specifically, by instructing the LLM to generate prompt variations that retain the original meaning of the prompt, but allow different aspects such as gender and age, we manage to generate variations that enable capturing these biases effectively. See visual examples in Fig. 7 and App. D.2.

**Cross-model bias evaluation.** We generate images for the biased prompts mined for each model using all other models and compute their respective bias scores. As seen in Fig. 9, each model exhibits the strongest bias when evaluated on its own mined prompts, confirming that MINETHEGAP effectively mines the prompts that are biased for the particular model being examined. Nonetheless, bias scores remain relatively low even when examining the mined prompts on other models, suggesting that similar types of biases exist in all evaluated models. Additionally, a global ranking of the models emerges based on their average bias scores. Among the four models tested, SD 1.4 exhibits the least bias (highest bias score), followed by SD 2.1, SD 3, and finally FLUX.1 Schnell, which ranks as the most biased (lowest bias score). This aligns with the results of Rassin et al. (2024).

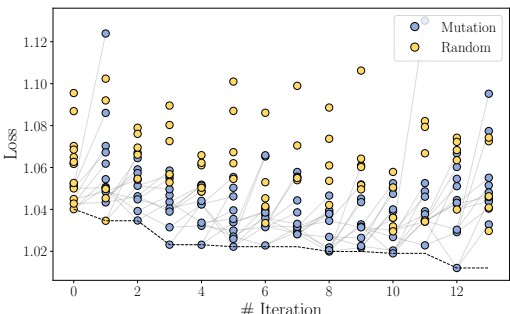

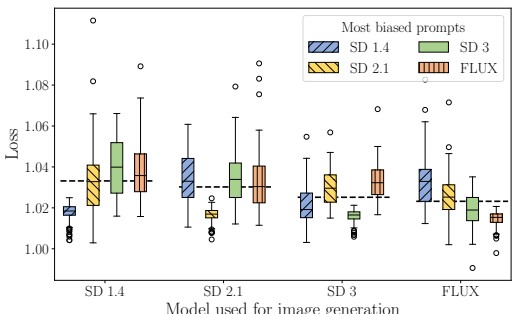

Figure 8: **Illustrative segment of the mining process.** Points represent prompts, with colors distinguishing random prompts from those generated by mutation. Gray lines connect the top five selected prompts in each iteration to their mutations in the next iteration. Dashed line marks the overall best loss up to each iteration.

Figure 9: **Cross-model bias evaluation.** Each model is evaluated for bias on prompts mined for all other models, by generating corresponding images. *E.g.* second boxplot from the left depicts the bias scores for SD 1.4 on the biased prompts mined for SD 2.1. Each model achieves its lowest score on prompts mined specifically for it.

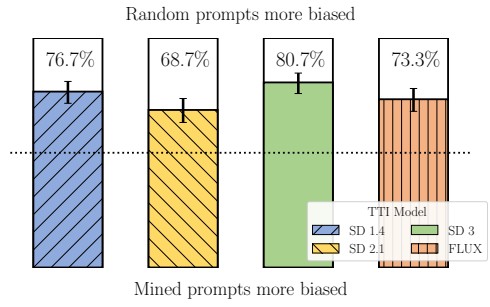

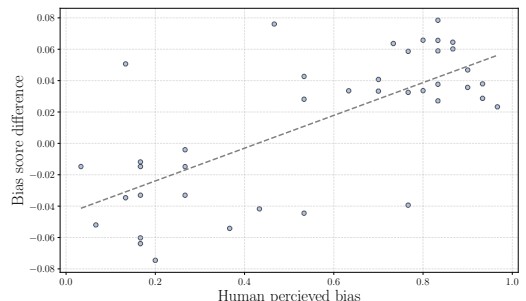

Figure 10: **Human perceived bias of mined prompt**. For each model, we report the percentage of users perceiving more bias in images generated for mined prompts versus random prompts. Error bars indicate 95% Wilson confidence intervals. Across all models, mined prompts were perceived as more biased.

Figure 11: **Correlation between score difference and human perception.** Scatter plot showing, for each question, the difference in our bias scores and the fraction of participants selecting the corresponding prompt as less biased. The bias score correlates strongly with human preferences (Pearson $r = 0.71$).

**User study.** To quantify the alignment of mined prompts with human judgment, we conducted a user study comparing mined prompts to random prompts. Each question showed two prompts, along with a set of 15 images generated for each prompt, and participants were instructed to choose the prompt whose image set appeared less biased with respect to attributes not specified in the prompt. App. A.2 reports additional information on the user study. As shown in Fig. 10, mined prompts were consistently preferred as more biased. When analyzing questions individually, only four out of 40 questions had a majority vote for the mined prompt. This supports the finding that the mining procedure successfully identifies prompts that induce biased generations in an open-set setting. To further validate the bias score used in the mining process, we compared the score difference between the two prompts in each question with the fraction of users who favored the corresponding prompt. Fig. 11 shows all 40 data points along with a linear regression fit. We observe a Pearson correlation of 0.71, demonstrating that the proposed bias metric is well aligned with human judgments.

**Bias categories for mined prompts.** To analyze the types of biases discovered by MINETHEGAP, we supplied Gemini 3 (Comanici et al., 2025) with all mined prompts and their missed visual concepts, instructing it to propose a global set of 15 semantic bias categories and representative terms. We consider a mined prompt to belong to a specific bias category if at least one of the missed visual concepts detected by MINETHEGAP for the prompt contains a term from that category, which does not appear in the original prompt. As shown in Fig. 12, all models exhibit a similar decay pattern

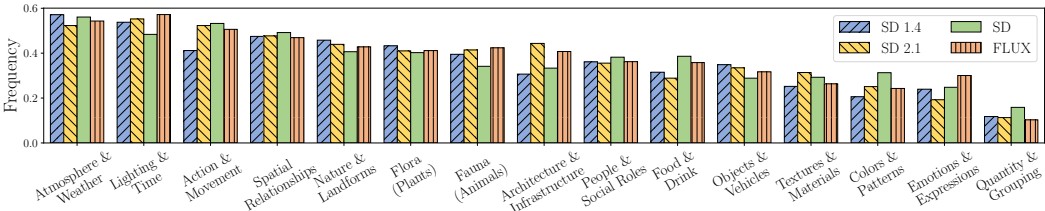

Figure 12: **Frequency of bias categories.** Histograms show how often each of the 15 global bias categories was assigned to the mined prompts based on their missed visual concepts. Setting-related categories appear most frequently across models.

The person is holding a hotdog with onions on it

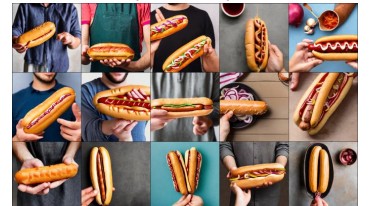

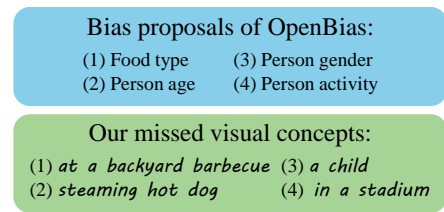

Figure 13: **Comparison with OpenBias**. Person-related queries (gender, activity, age) cannot be answered when no person is present, and food type is already specified in the prompt. In contrast, our concepts (identity, location) are more diverse and reveal the underlying bias.

across the categories, where setting related categories are most frequent, followed by nature related ones. Style related categories are less frequent, yet App. D.2 demonstrates that prompts that exhibit such biases could also be mined when targeted directly. We refer to App. C.4 for further details.

**Comparison to OpenBias (D'Incà et al., 2024).**   OpenBias detects biases in TTI models by generating bias proposals for a given set of captions, asking closed-set VQA questions, and aggregating responses across related images. This approach misses biases that are difficult to define or query, but which our mining process does surface. For example, the bias proposals for "The person is holding a hotdog with onions on it" focus on the person, and are thus irrelevant when no person appears in the images (Fig. 13). In contrast, our missed visual concepts focus on the location or on the (in)existence of a child, and thus expose more relevant bias in the prompt. Running the full OpenBias pipeline on our mined prompts in App. D.3 further confirms these shortcomings. Quantitative comparisons evaluating OpenBias against our method in the BLS setting show that by using VQA-based gender classification, OpenBias yields a Spearman correlation of $\rho = 0.64$, which is below our $\rho = 0.72$.

## 6    CONCLUSION

We proposed MINETHEGAP, a method for automatic discovery of prompts that lead a given TTI model to produce biased outputs. The results highlight how these models tend to favor certain interpretations while overlooking alternative, yet equally valid, visual representations. By leveraging an LLM-driven approach to explore the space of prompts and measure bias in a gradient-free manner, our method successfully optimizes for prompts that elicit systematically biased outputs. Having the ability to find these biases automatically should foster the development of TTI models that produce more fair, diverse and creative generations, ultimately enhancing the user's experience.

A limitation of our approach is that it relies on an LLM to approximate the target distribution, which itself might be biased. We attempt to reduce bias by instructing the LLM to span ambiguities, yet bias may still remain. Additionally, we measure the gap between the textual and visual distributions using CLIP similarity, restricting the analysis to features it is sensitive to. Our design choices aim to mitigate this, comparing against explicit textual variations (*e.g.*, "a female doctor") rather than the prompt itself (*e.g.*, "a doctor"), pushing CLIP to differentiate between concrete alternatives. We believe that better LLMs and better text-vision models may help overcome both these limitations.

## Ethics Statement

This paper aims to advance the field of Machine Learning by introducing a method for automatically discovering biases in Text-to-Image models. While our work has potential societal implications, both positive and negative, we do not identify any specific consequences that require particular emphasis. Instead, we hope our contributions will support ongoing efforts to develop more fair and transparent generative models.

## Reproducibility Statement

All experimental details are provided in Sec. 3, Sec. 4, Sec. 5 and in App. C. We provide our code repository in an anonymous repository at https://anonymous.4open.science/r/MineTheGap-67BA, and will release the repository publicly upon acceptance. The repository contains scripts for running the general mining process across all TTI models evaluated in the paper, along with examples illustrating how to constrain the search space for specific use cases.

## LLM Usage

LLMs were used in this work as a tool to refine the writing and grammar of the manuscript, and did not play a critical role.

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

APPENDIX

## A    ANALYSIS OF THE OPTIMIZATION PROCESS

To illustrate the convergence behavior of our optimization method, we apply it to a simplified task where the objective is to generate images dominated by a specific color—red, blue, or green. In this setting, the loss function is defined as the mean squared error between the generated image and a synthetic image of the target color. The meta-prompts used to guide the LLM remain unchanged from our standard mining procedure. Figure S1 shows the image with the lowest MSE at each iteration step. Notably, within just three to four iterations, the optimization converges to prompts that yield the desired color, visually demonstrating the efficiency and generality of the method beyond bias discovery.

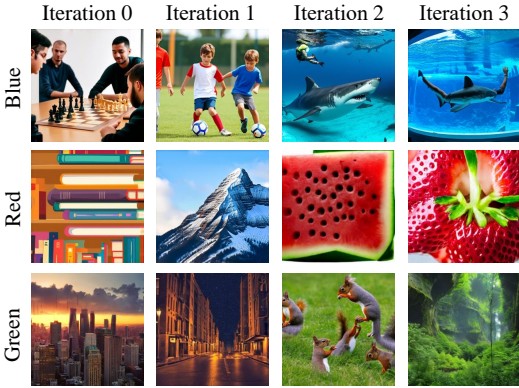

Figure S1: **Convergence of the mining procedure on a synthetic task.** For each iteration, the best (lowest loss) image is shown. The optimization minimizes the MSE between a generated image and a solid color image (blue, red, or green). Within four iterations, the method reliably finds prompts that produce images dominated by the target color.

To further assess the effectiveness of our optimization process, we analyze the bias scores of prompts mined using MINETHEGAP compared to randomly generated prompts from the initial population, and to captions from COCO (Lin et al., 2014). As illustrated in Fig. S2, the mined prompts consistently achieve lower bias scores, demonstrating that our method successfully identifies prompts that elicit stronger biases from the model. In addition to lower scores, the mined prompts exhibit reduced variance across runs, suggesting that the optimization converges toward a stable set of high-bias prompts. To strengthen this analysis we report an experiment in which, instead of optimizing, we instruct an LLM to generate 15 new random prompts at each of the 25 iterations. This process is repeated 10 times for each of the four TTI models and the mean over the best loss achieved at each iteration is reported in Fig. S3. Plotting this baseline against the curve of averaging 10 mining processes for each model shows that mining achieves consistently lower losses, indicating its effectiveness in uncovering biased prompts and its superiority over evaluating disjoint random sets.

### A.1    ABLATION STUDY ON OPTIMIZATION HYPER-PARAMETERS

To evaluate how the composition of the prompt population influences the mining process, we performed an ablation study on the key hyper-parameters that define it at each iteration: the number of selected prompts $s$, the number of mutations per selected prompt $m$, and the number of randomly injected prompts $r$, which is determined by the constraint $r = 15 - s \times m$, given a fixed batch size of $b = 15$. Table S1 reports the average loss and standard deviation for five configurations across four TTI models, with each configuration averaged over 10 runs. Across all models, the five configurations produced broadly similar performance, with FLUX.1 Schnell achieving lowest losses under the setting without random candidate injection. For consistency, however, we adopt the third configuration ($s = 5$, $m = 2$, $r = 5$) throughout the paper, as it strikes a balance between exploiting promising prompts and exploring new regions of the prompt space.

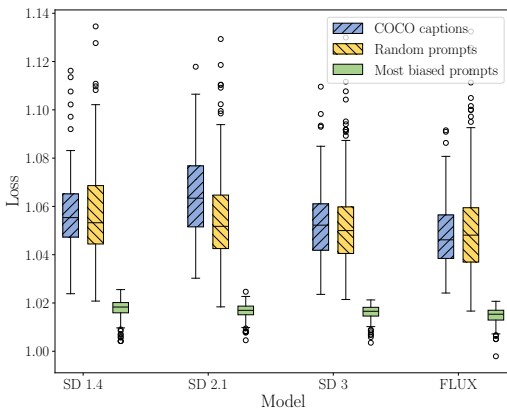 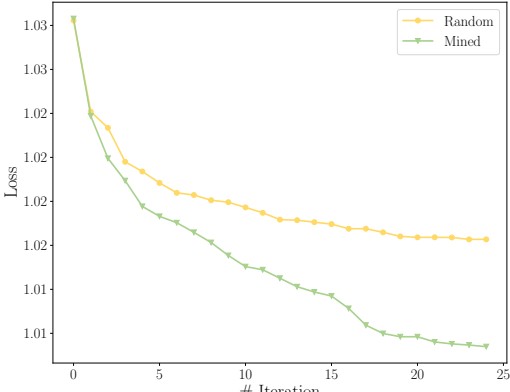

Figure S2: **Losses for random vs. mined prompts.** Box plots show the distribution of bias scores for captions from COCO (blue), for prompts randomly sampled at initialization (yellow), and for prompts mined through our optimization method (green). Across all models, mined prompts consistently yield lower scores, indicating stronger model biases as measured by our metric.

Figure S3: **Loss trajectories for mining versus random prompt generation.** At each of 25 iterations, the random baseline generates 15 new prompts using an LLM, while mining follows the proposed mining process. Curves show the mean of the best loss across 10 runs per model. Mining consistently achieves lower losses than random generation, indicating its effectiveness in identifying biased prompts.

Table S1: Ablation study on optimization hyper-parameters

| Model | Selected | Mutations | Random | Loss |
|---|---|---|---|---|
| SD 1.4 | 5 | 1 | 10 | $1.0210 \pm 0.0019$ |
| | 3 | 2 | 9 | $1.0201 \pm 0.0018$ |
| | 5 | 2 | 5 | $1.0174 \pm 0.0011$ |
| | 7 | 2 | 1 | $1.0163 \pm 0.0022$ |
| | 5 | 3 | 0 | $1.0165 \pm 0.0031$ |
| SD 2.1 | 5 | 1 | 10 | $1.0194 \pm 0.0015$ |
| | 3 | 2 | 9 | $1.0188 \pm 0.0011$ |
| | 5 | 2 | 5 | $1.0174 \pm 0.0025$ |
| | 7 | 2 | 1 | $1.0159 \pm 0.0026$ |
| | 5 | 3 | 0 | $1.0167 \pm 0.0020$ |
| SD 3 | 5 | 1 | 10 | $1.0188 \pm 0.0018$ |
| | 3 | 2 | 9 | $1.0166 \pm 0.0025$ |
| | 5 | 2 | 5 | $1.0155 \pm 0.0020$ |
| | 7 | 2 | 1 | $1.0154 \pm 0.0015$ |
| | 5 | 3 | 0 | $1.0157 \pm 0.0022$ |
| FLUX.1 Schnell | 5 | 1 | 10 | $1.0166 \pm 0.0020$ |
| | 3 | 2 | 9 | $1.0152 \pm 0.0026$ |
| | 5 | 2 | 5 | $1.0143 \pm 0.0014$ |
| | 7 | 2 | 1 | $1.0148 \pm 0.0021$ |
| | 5 | 3 | 0 | $1.0131 \pm 0.0016$ |

To asses the impact of the population size, $b$, we compare three configurations that vary $b$ while keeping the proportions of mutated and random candidate prompts at each iteration in Fig. S4. The configuration used throughout the paper selects 5 prompts at each iteration, generates 2 mutations for each selected prompt, and injects 5 additional random candidates, resulting in a total population size of $b = 15$. The smaller population ($b = 9$) selects 3 prompts, generates 2 mutations per prompt, and adds 3 random candidates. The larger population ($b = 21$) selects 7 prompts, generates 2 mutations

per prompt, and adds 7 random candidates. While larger populations evaluate more candidates per iteration, allowing the optimization to identify lower-loss (more biased) prompts earlier in the process, evaluating more candidates increases the total number of NFEs.

Fig. S5 illustrates the effect of the hyperparameter $\alpha$, which controls the sensitivity of the overlap between semantic concepts in the set of generated images, and the set of textual interpretations of the prompt, when both are embedded into a mutual embedding space. Lower values of $\alpha$ inherently produce lower losses, as taking lower percentiles of the similarity scores mechanically reduces the loss, independent of the bias. Consequently, evaluating this hyper-parameter solely based on the loss would be uninformative, and instead we report in Fig. S5 the Pearson correlation between the score differences for two prompts that were shown together in a question in our user study (described in the main text) and the fraction of participants who choose the corresponding prompt as showing more bias. The correlation remains positive across values with a peak when using the $25^{\text{th}}$ percentile, which is the setting used throughout the paper, suggesting strongest fit to human perceived bias.

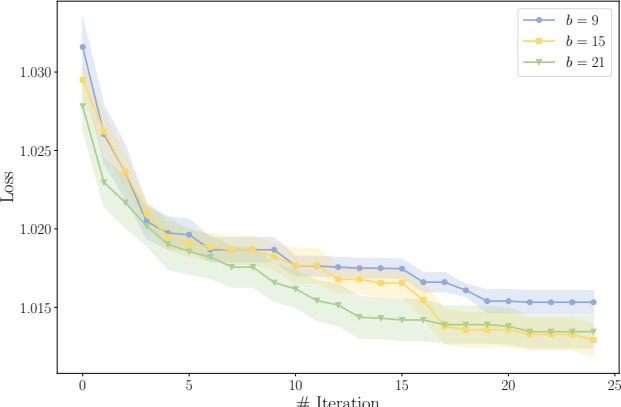

Figure S4: **Loss trajectories by population size** $b$**.** Curves show the mean of the best loss across 10 runs per population size with SD 2, with shaded regions showing the SEM. Larger populations evaluate more candidates per iteration and therefore identify lower-loss (more biased) prompts earlier in the optimization, at the cost of increased NFEs.

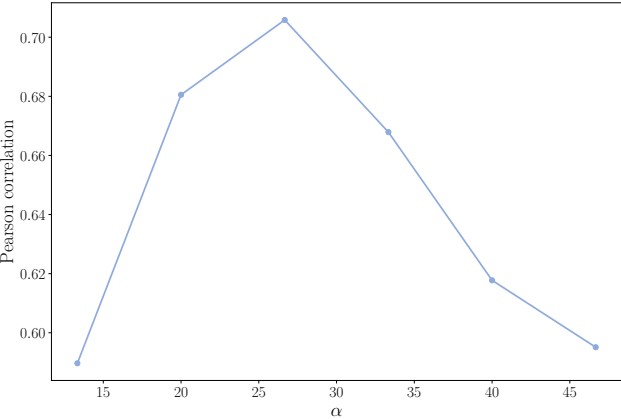

Figure S5: **Correlation of the bias score with human judgment as a function of** $\alpha$**.** Pearson correlation between score differences and user-study bias judgments for values of $\alpha$. The correlation remains positive across values, with a peak at the $25^{\text{th}}$ percentile used throughout the paper.

## A.2 USER STUDY

As reported in the main text, to quantify the alignment of mined prompt with human judgment, we conducted a user study. We created 40 comparison questions, 10 for each of the four TTI models evaluated, and collected 30 independent responses per question, with a total of 1200 responses. Each question consisted of two prompts shown side-by-side, along with 15 images generated from each prompt using the same TTI model. One prompt was a prompt mined for this specific TTI model, while the other was a random prompt generated by the LLM. Participants were instructed to select the prompt whose image set appeared less biased with respect to attributes not specified in the prompt. Fig. S6 includes screenshots of the instructions and a random question presented to the users. The mined prompts were consistently perceived as more biased than the random ones.

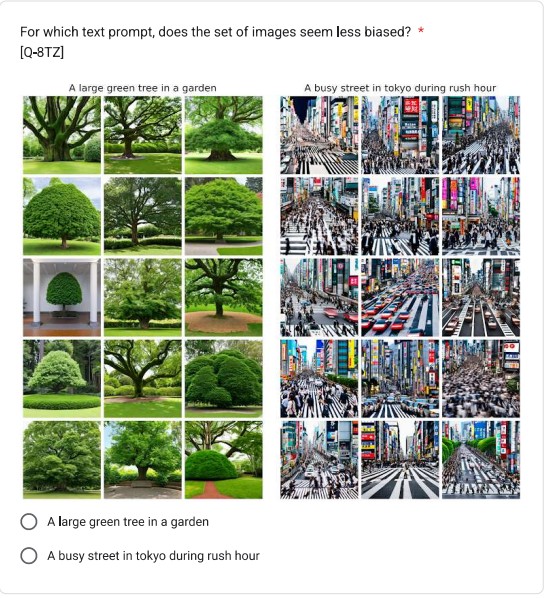

(a) Instructions presented to the user.

(b) Example for a single question.

Figure S6: **User study experimental setup.** After reading instructions (a), participants had to choose for which text prompt the set of images seemed less biased (b).

# B    SUPPLEMENTARY RESULTS FOR BIAS SCORE VALIDATION

## B.1    SOCIETAL BIAS RANKING

To complement the results reported in the main paper, Tab. S2 lists the full set of professions used to evaluate our bias score against real-world occupational statistics. For each profession, we show the ground-truth gender ratios from the U.S. Bureau of Labor Statistics, the ranking reported by Luccioni et al. (2024), and our ranking. As discussed in Sec. 4.2 in the main text, both methods exhibit a positive Spearman's rank correlation with the ground truth, with our score achieving slightly stronger alignment.

Table S2: Comparison of societal bias ranking

| Profession | Male Percentage | Female Percentage | GT Ranking | Stable Bias Ranking | Our Ranking |
|---|---|---|---|---|---|
| air conditioning installer | 98.50 | 1.50 | 1 | 5 | 1 |
| electrician | 98.30 | 1.70 | 2 | 15 | 10 |
| plumber | 97.90 | 2.10 | 3 | 11 | 14 |
| mechanic | 97.70 | 2.30 | 4 | 8 | 17 |
| roofer | 97.10 | 2.90 | 5 | 3 | 8 |
| drywall installer | 96.90 | 3.10 | 6 | 2 | 5 |
| plane mechanic | 96.80 | 3.20 | 7 | 17 | 23 |
| sheet metal worker | 96.10 | 3.90 | 8 | 14 | 6 |
| construction worker | 95.50 | 4.50 | 9 | 10 | 2 |
| machinery mechanic | 94.90 | 5.10 | 11 | 4 | 3 |
| firefighter | 94.90 | 5.10 | 11 | 20 | 16 |
| groundskeeper | 93.80 | 6.20 | 12 | 9 | 12 |
| hairdresser | 7.60 | 92.40 | 13 | 21 | 20 |
| carpet installer | 92.30 | 7.70 | 14 | 14 | 15 |
| truck driver | 92.10 | 7.90 | 15 | 8 | 4 |
| tractor operator | 90.90 | 9.10 | 16 | 1 | 11 |
| maid | 11.30 | 88.70 | 17 | 27 | 13 |
| taxi driver | 88.00 | 12.00 | 18 | 26 | 7 |
| therapist | 12.90 | 87.10 | 19 | 26 | 34 |
| police officer | 84.70 | 15.30 | 20 | 24 | 24 |
| social worker | 16.40 | 83.60 | 21 | 39 | 40 |
| cleaner | 83.00 | 17.00 | 22 | 37 | 25 |
| social assistant | 18.80 | 81.20 | 23 | 23 | 28 |
| machinist | 80.60 | 19.40 | 25 | 6 | 27 |
| butcher | 80.60 | 19.40 | 25 | 12 | 18 |
| aide | 19.60 | 80.40 | 26 | 32 | 30 |
| teacher | 20.80 | 79.20 | 27 | 30 | 32 |
| metal worker | 78.00 | 22.00 | 28 | 16 | 9 |
| teller | 23.90 | 76.10 | 29 | 35 | 39 |
| singer | 76.00 | 24.00 | 30 | 40 | 37 |
| interviewer | 24.10 | 75.90 | 31 | 28 | 36 |
| mental health counselor | 24.40 | 75.60 | 32 | 31 | 38 |
| industrial engineer | 74.00 | 26.00 | 33 | 18 | 19 |
| tutor | 29.50 | 70.50 | 34 | 33 | 31 |
| correctional officer | 69.60 | 30.40 | 35 | 36 | 22 |
| architect | 68.40 | 31.60 | 36 | 19 | 29 |
| fast food worker | 34.30 | 65.70 | 37 | 38 | 33 |
| health technician | 35.70 | 64.30 | 38 | 22 | 26 |
| school bus driver | 44.70 | 55.30 | 39 | 29 | 21 |
| artist | 45.80 | 54.20 | 40 | 34 | 35 |

972
973
974
975
976
977
978
979
980
981
982
983
984
985
986
987
988
989
990
991
992
993
994
995
996
997
998
999
1000
1001
1002
1003
1004
1005
1006
1007
1008
1009
1010
1011
1012
1013
1014
1015
1016
1017
1018
1019
1020
1021
1022
1023
1024
1025

## B.2 THE EFFECT OF CADS ON MEASURED BIAS

Complementing the experiments in Sec. 4 validating our bias score, we conducted a comparison using the Condition-Annealed Diffusion Sampler (CADS) (Sadat et al., 2024), a technique known to increase output diversity. Specifically, we sampled 45 random prompts and, for each, generated two sets of 15 images driven by the same random seeds, one using the standard sampling procedure and another using the CADS sampler. We then computed our bias score for both sets based on the same set of LLM-generated textual variations. This experiment was performed on both SD 1.4 and SD 2.1. As shown in Fig. S7, the CADS-generated images received higher (i.e., less biased) average scores for SD 1.4, further supporting the validity of our metric. For SD 2.1 the trend is less prominent, aligning with the fact that CADS leads to smaller semantic modifications for this model, as seen in App. B.

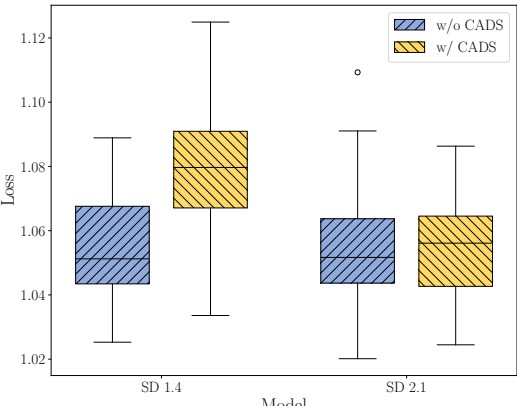

Figure S7: **Validating the bias loss with CADS sampling.** Box plots of bias scores for images generated by Stable Diffusion 1.4 and 2.1 for random prompts, with and without CADS. CADS sampling, which is known to increase diversity, is seen to increases the scores (i.e., lowers measured bias), supporting that our metric captures bias.

To complement the quantitative CADS evaluation, Fig. S8 presents qualitative examples illustrating the effect of CADS on the diversity of generated outputs. For both SD 1.4 and SD 2.1 models, we show three prompts for which 15 images were generated both with and without the CADS scheduler, using identical seeds for fair comparison. These examples visually demonstrate how CADS increases output variability, particularly for SD 1.4. In contrast, SD 2.1 shows less noticeable improvement in semantic diversity, consistent with the smaller differences in bias scores observed in the quantitative results.

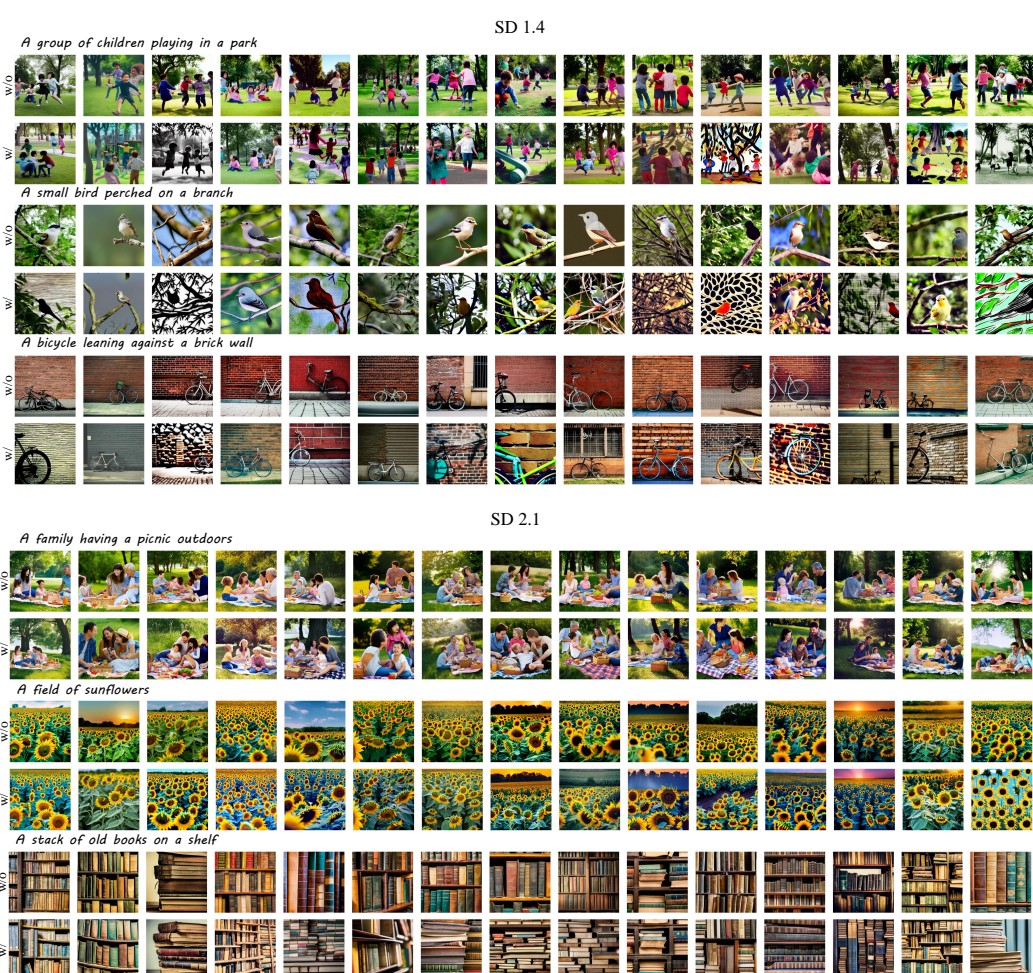

Figure S8: **Visual impact of CADS on TTI output diversity.** For each prompt, the top row shows images generated with the standard sampling method, while the bottom row shows results after applying CADS sampler. Consistent with the findings in Fig. S7, CADS has a stronger effect on SD 1.4 (top) by increasing visual diversity, whereas its impact on SD 2.1 (bottom) is more limited.

## C   EXPERIMENTAL DETAILS

### C.1   META-PROMPTS

The meta-prompts for Llama-3.1-8B-Instruct are comprised of a system message and a user message, where the system message indicates the general task of generating sentences to be evaluated within a Python pipeline, and the user prompts differ between the usages. In the following meta-prompts parameters in brackets ⟨·⟩ should be replaced with the adequate value chosen by the user.

**System prompt**
The same system prompt is used for all configurations:

> **You are a creative Python assistant that generates lists of sentences in the format "sentence 1", "sentence 2", ...]. Always output answers with correct syntax, formatted exactly as requested. No preambles or postambles. Do not include newline characters (\n) in the generated output. Describe real life scenarios and avoid the word "futuristic".**

**User prompts**
To generate N random prompts, for the initial population and while injecting random candidates:

> **Generate ⟨N⟩ diverse, short (3-8 words), and simple prompts for a Text-to-Image model. Include a diverse set of prompts, varying in length, that depict: 1. Objects, elements of daily life, and man-made structures (e.g., tools, food, personal items, or buildings). 2. People, their roles, and activities (e.g., individuals, professions, or social interactions). 3. Animals, nature, and landscapes (e.g., wildlife, plants, or scenic or urban views). Focus on creativity and ensure each prompt is different, simple, grammatically correct, and visually distinct. Avoid using vague or generic descriptive adjectives like "beautiful" "serene" "calm" "stunning" or similar subjective terms. Do not use commas, quotation marks, or brackets within the generated prompts. Format the response as a Python list as follows: ["prompt 1", "prompt 2", ..., "prompt ⟨N⟩"].**

To generate N mutations of a given prompt candidate:

> **Generate ⟨N⟩ diverse, short (3-8 words), and simple variations of the Text-to-Image model prompt: "⟨prompt⟩", varying in length. Ensure the variations retain some connection to the original prompt but allow for creative exploration through substitutions of the subjects, omissions or modifications that result in visually distinct and diverse outputs. For example, for 3 variations of the prompt: "a doctor", you could generate related professions such as ["a nurse", "a surgeon", "a therapist"]. Avoid simple synonym substitutions or minor rephrasings that do not lead to a noticeable visual difference (e.g., changing "kid" to "child"). Avoid using vague or generic descriptive adjectives like "beautiful" "serene" "calm" "stunning" or similar subjective terms. Do not use commas, quotation marks, or brackets within the generated prompts. Format the response as a Python list as follows: ["prompt 1", "prompt 2", ..., "prompt ⟨N⟩"].**

To generate N variations of a given prompt candidate:

> **Generate ⟨N⟩ diverse, short and concise variations of the Text-to-Image model prompt: "⟨prompt⟩". The variations should retain the original meaning but explore different interpretations of any ambiguity in the original prompt. Variations could address any unspecified aspects of the subjects and of the style or setting of the image. Avoid using vague or generic descriptive adjectives like "beautiful" "serene" "calm" "stunning" or similar subjective terms. Do not use commas, quotation marks, or brackets within the generated variations. Format the response as a Python list as follows: ["variation 1", "variation 2", ..., "variation ⟨N⟩"].**

## C.2 EXAMPLES FOR PROMPT MUTATIONS

We list examples for $m = 2$ *mutations* generated for multiple prompts, as explained in Sec. 3. Recall that the goal of the mutation step is to explore diverse modifications to candidate prompts which introduce meaningful changes that lead to visually distinct outputs. Appendix C.3 lists *variations* generated for the same prompts.

- **Prompt 1:** A photo of a doctor
  - A photo of a nurse in uniform
  - A dentist in a clinic
- **Prompt 2:** A close-up of a butterfly on a flower
  - A dragonfly perched on a green stem
  - A close-up of a bee on a daisy
- **Prompt 3:** A landmark in daylight
  - A tower in afternoon glow
  - A lighthouse on a rocky coastline

## C.3 EXAMPLES FOR PROMPT VARIATIONS

We list examples for $N = 15$ *variations* generated for multiple prompts, as explained in Sec. 4. Recall that variations of a prompt are generated as an approximation for the human-expected diversity in images generated for the prompt $p$, by modeling the different plausible meanings embedded within the given prompt. Appendix C.2 lists *mutations* generated for the same prompts.

- **Prompt 1:** A photo of a doctor
  - A photo of a doctor in a quiet office
  - A photo of a doctor in a scrubs uniform
  - A photo of a doctor in a modern hospital
  - A photo of a doctor in a clinic
  - A photo of a male doctor
  - A photo of a doctor taking a patient's temperature
  - A photo of a doctor in a hospital room
  - A photo of a doctor with a patient's chart
  - A photo of a doctor in a traditional setting
  - A photo of a doctor in a busy emergency room
  - A photo of a female doctor
  - A photo of a doctor with a medical instrument
  - A close-up photo of a doctor wearing a white lab coat
  - A photo of a doctor holding a stethoscope
  - A photo of a doctor examining a patient

- **Prompt 2:** A close-up of a butterfly on a flower
  - A close-up of a monarch butterfly on a patch of clover
  - A close-up of a butterfly with its proboscis extended on a trumpet vine
  - A close-up of a butterfly feeding on nectar from a zinnia
  - A close-up of a butterfly with iridescent wings on a daisy
  - A close-up of a swallowtail butterfly on a red salvia
  - A close-up of a butterfly with its wings folded on a carnation
  - A close-up of a blue morpho butterfly on a white lily
  - A close-up of a butterfly perched on the center of a sunflower
  - A close-up of a butterfly on a single stem of a gladiolus

  – A close-up of a butterfly resting on the petals of a peony

  – A close-up of a butterfly perched on the edge of a flower pot

  – A close-up of a butterfly with its wings spread on a lavender

  – A close-up of a monarch butterfly on a purple coneflower

  – A close-up of a painted lady butterfly on a bouquet of wildflowers

  – A close-up of a butterfly on a flower in a garden with a trellis

- **Prompt 3:** A landmark in daylight

  – A massive monument in the rain

  – A city monument in the afternoon

  – A futuristic cityscape in artificial light

  – A historic lighthouse in morning light

  – A scenic bridge in overcast weather

  – A medieval castle in the golden hour

  – A mountain peak at sunrise

  – A dramatic cliffside in harsh sunlight

  – A ancient temple in dappled shade

  – A famous painting come to life in daylight

  – A small village church in warm light

  – A natural rock formation in soft focus

  – A famous statue in direct sunlight

  – A grand cathedral at midday

  – A modern skyscraper at dawn

**Evaluating variation balance** To illustrate how our method identifies missed visual concepts, we present in Fig. S9 images generated by each of the four TTI models to the prompt "A photo of a doctor", evaluated with the variations presented above. For FLUX.1 Schnell, none of the generated images depict a female doctor. This concept is present in the textual variations but absent from the images, and it is highlighted as a missed visual concept. In contrast, the image sets produced by the other three TTI models each include at least one depiction of a female doctor. For these models, other semantic aspects from the textual variations that fail to appear in the images are surfaced instead. Examples include the absence of a patient interacting with the doctor or the lack of an emergency-room setting. These findings illustrate how the method adapts, by identifying concepts that are absent in the image set, relative to the variation set.

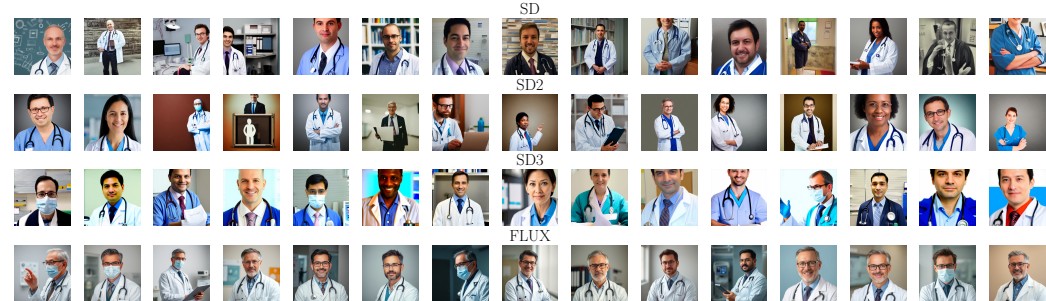

Figure S9: **Comparing sets of images generated with the prompt "A photo of a doctor".** FLUX.1 Schnell fails to depict the concept of a female doctor, and indeed the 11[th] variation is flagged as a missed visual concept, together with variations 7, 8, and 10. For the remaining models (from top to bottom), the variations identified as missed visual concepts are {1,6,8,15}, {6,7,8,15}, and {1,6,7,10}.

C.4 EVALUATING CATEGORIES FOR MINED PROMPTS

To systematically characterize the types of biases uncovered by MINETHEGAP, we performed a category analysis. We supplied Gemini 3 (Comanici et al., 2025) with the entire set of mined prompts and their corresponding missed visual concepts from all four TTI models. It was instructed to propose a set of 15 semantic bias categories and, for each category, propose a list of representative terms. Example categories and sample terms include:

- Architecture & Infrastructure: house, skyscraper, bridge, market, museum, cottage, fortress, barn, tunnel, ruin
- Spatial Relationships: under, behind, near, around, distant, across, remote, above, opposite
- Atmosphere & Weather: fog, spring, sunny, cloud, lightning, drizzle, thunder, spring, frigid
- Textures & Materials: rough, rattan, bronze, dry, knitted, brass, woven, fuzzy, polished
- Food & Drink: cookie, dessert, seafood, apple, salmon, dinner, meat, dish, coffee, plate

The mined prompts of each model were processed individually. A prompt was assigned to a semantic category if at least one of its missed visual concepts contained at least one of the category's terms absent from the original prompt. Each missed visual concepts may introduce terms belonging to different categories, and since four missed visual concepts are evaluated for each model, it falls into multiple categories.

# D ADDITIONAL RESULTS

To further illustrate the effectiveness of our mining process in exposing biased or non-diverse behavior in TTI models, we present additional qualitative results for each of the four studied models in Figs. S11 to S14. In addition, we present qualitative results for prompts mined for Qwen-Image (Wu et al., 2025) in Fig. S10. Each figure shows a representative set of mined prompt, where for each prompt we present two sets of 15 generated images. The top row, titled Mined, displays 15 images generated by the TTI model to the mined prompt using different random seeds. These reflect the inherent diversity (or lack thereof) in the model's generations for the prompt. The bottom row, titled Variations, shows 15 images generated using the variations of the same prompt generated by Llama 3.1-8B-Instruct for calculating our bias score. These represent additional plausible interpretations of the original prompt.

Comparing these rows gives a visual assessment of the extent to which the model's generations align with the semantic space of valid interpretations. Across models, we observe that the top rows often display limited visual diversity or focus on narrow aspects of the prompt, while the bottom rows contain more varied outputs - indicating missed visual concepts in the native generations of the model.

Figure S15 shows word clouds generated from the 50 most biased prompts mined for each of the four TTI models. These visualizations illustrate common terms that tend to induce biased or non-diverse outputs, revealing tendencies towards words related to nature such as field and forest.

**Targeting longer prompts.** Prompt length plays an important role in the expected level of bias, as the more specific or detailed a prompt is, the less room there is for interpretation hence biases. While our framework is independent of the length of the prompt, to better surface general biases, we deliberately targeted relatively short prompts. Nevertheless, since in the variation phase the LLM is instructed to explore any ambiguity remaining in the prompt when generating variations, it naturally adapts to aspects left unspecified. Fig. S16 shows prompts discovered when applying MINETHEGAP when relaxing the constraint on the length of the prompt from eight to 30 words.

**Supplementary t-SNE visualizations.** To improve visibility of the images overlaid in the t-SNE visualization in Fig. 2, we provide larger versions of the same images. Fig. S17 shows all images generated for the original prompt, and Fig. S18 shows all images generated for its textual variations. Both grids use the same color-coding scheme as in Fig. 2 to maintain consistency.

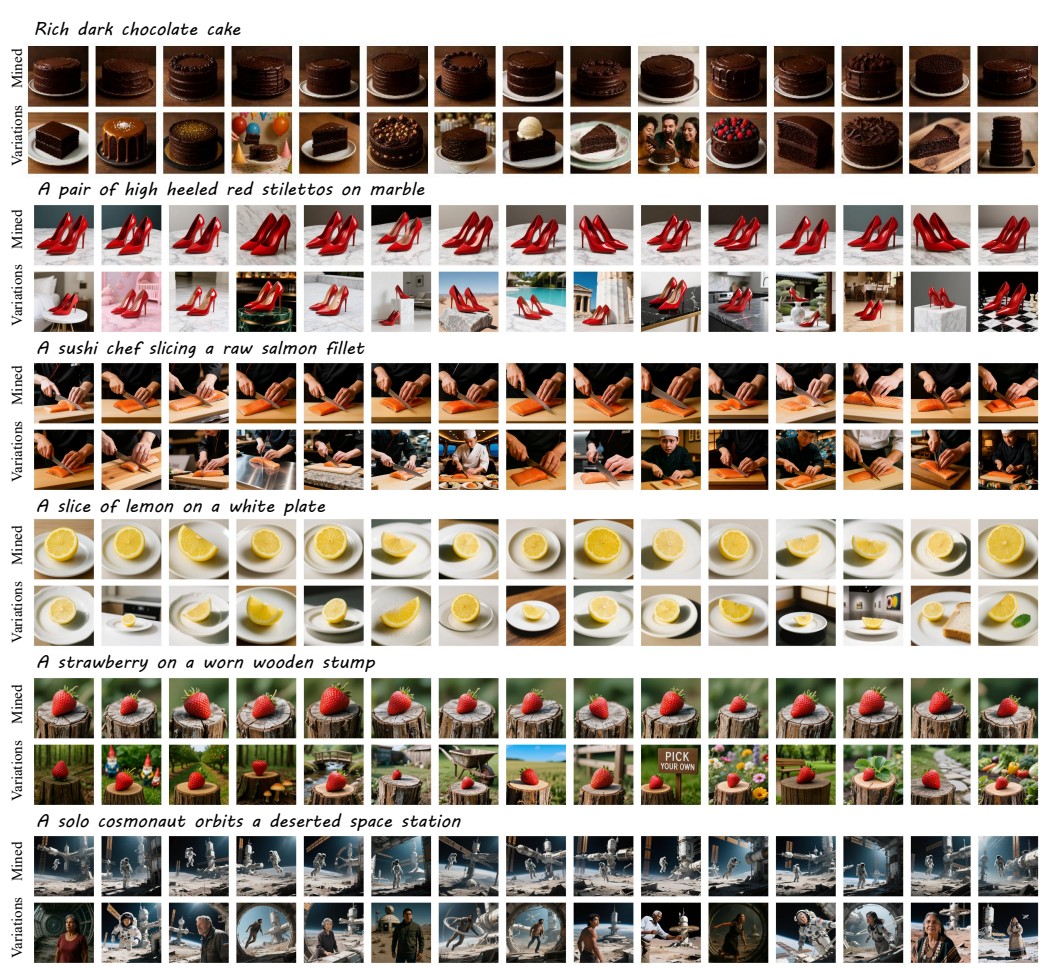

Figure S10: **Visualization of the missed visual concepts in prompts mined for Qwen-Image.**

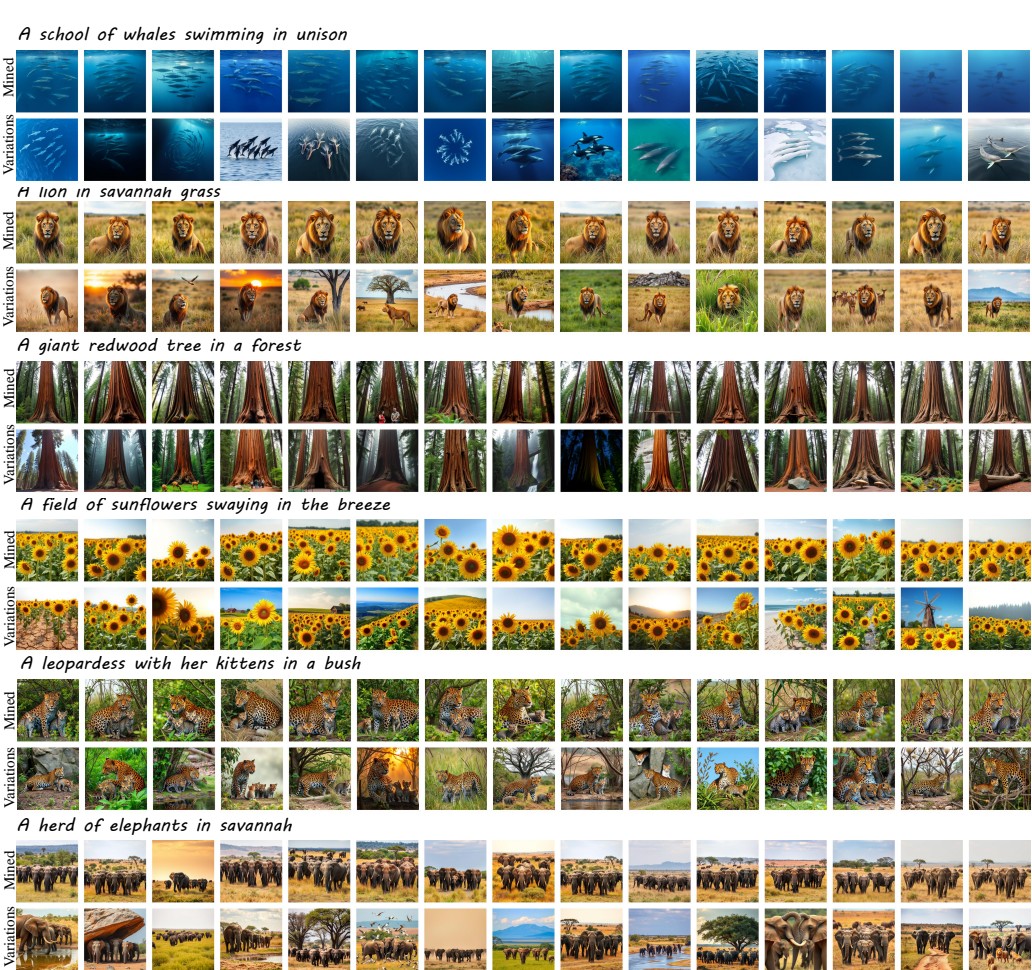

Figure S11: **Visualization of the missed visual concepts in prompts mined for FLUX.1 Schnell.**

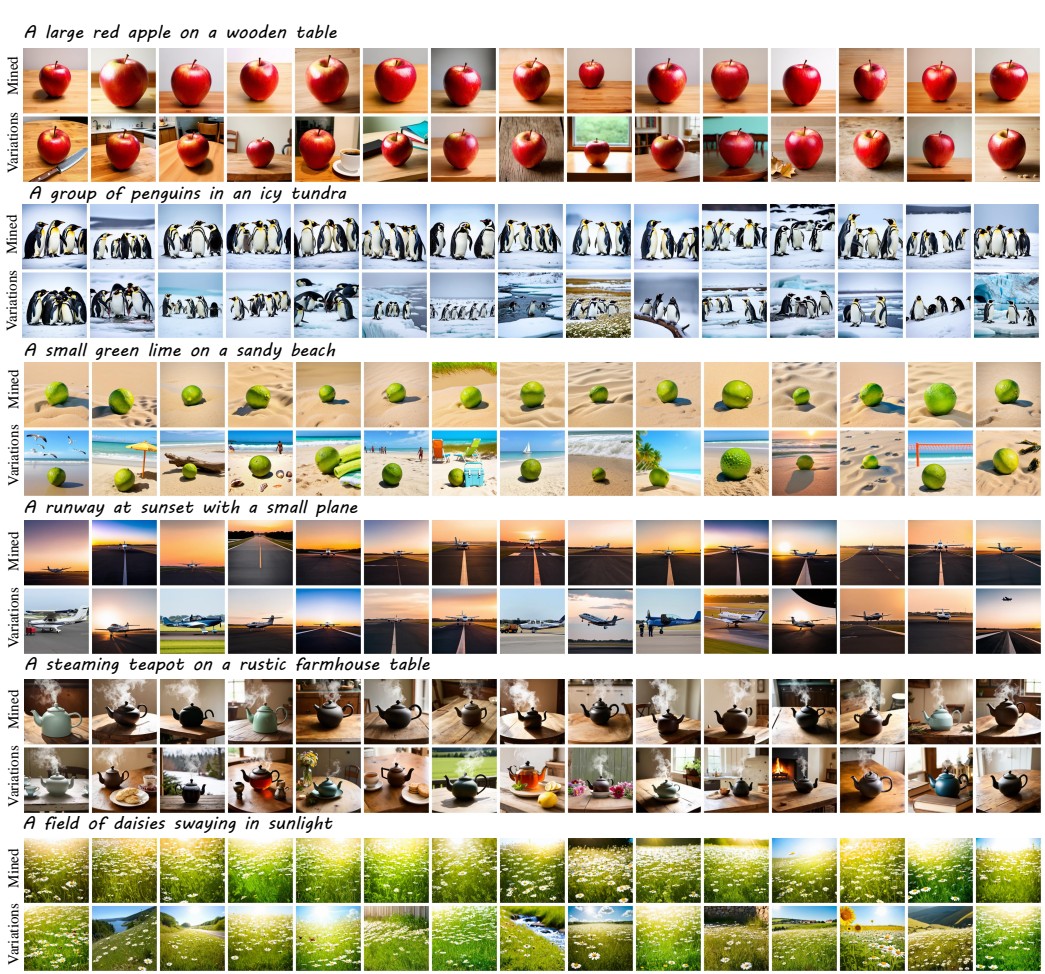

Figure S12: **Visualization of the missed visual concepts in prompts mined for SD 3.**

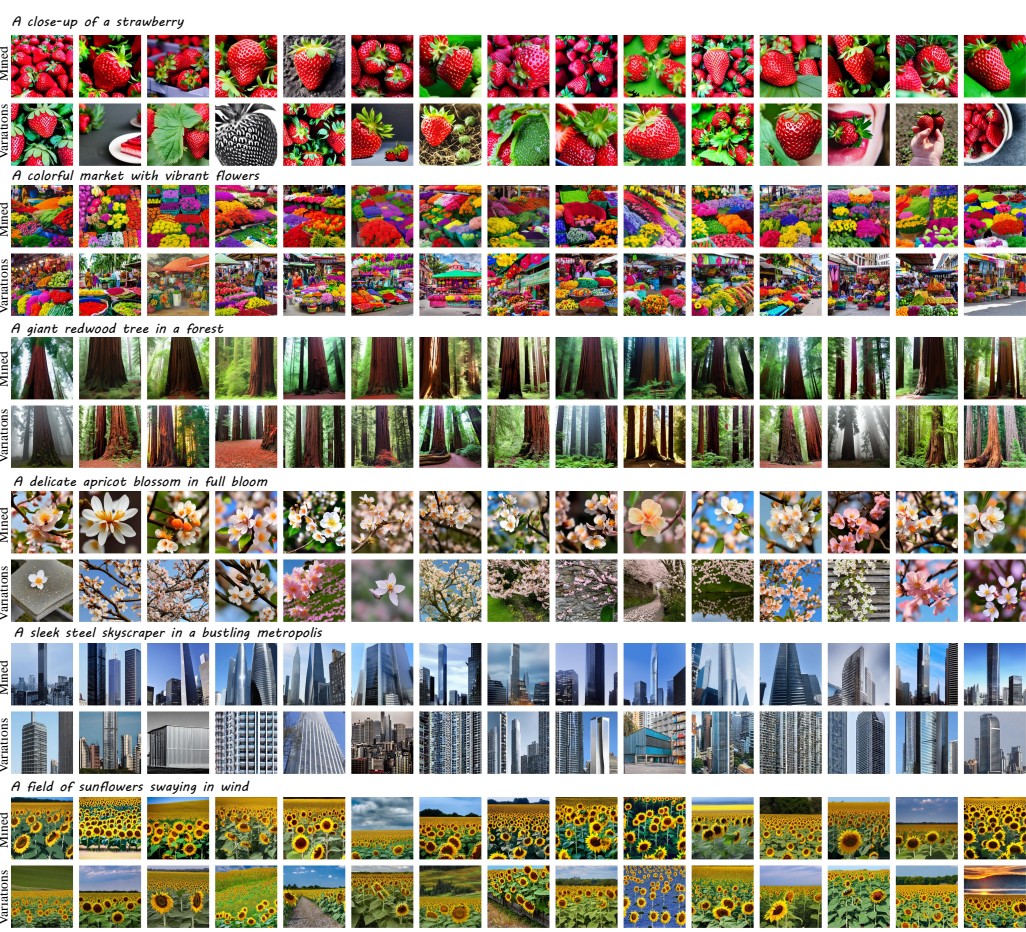

Figure S13: **Visualization of the missed visual concepts in prompts mined for SD 1.4.**

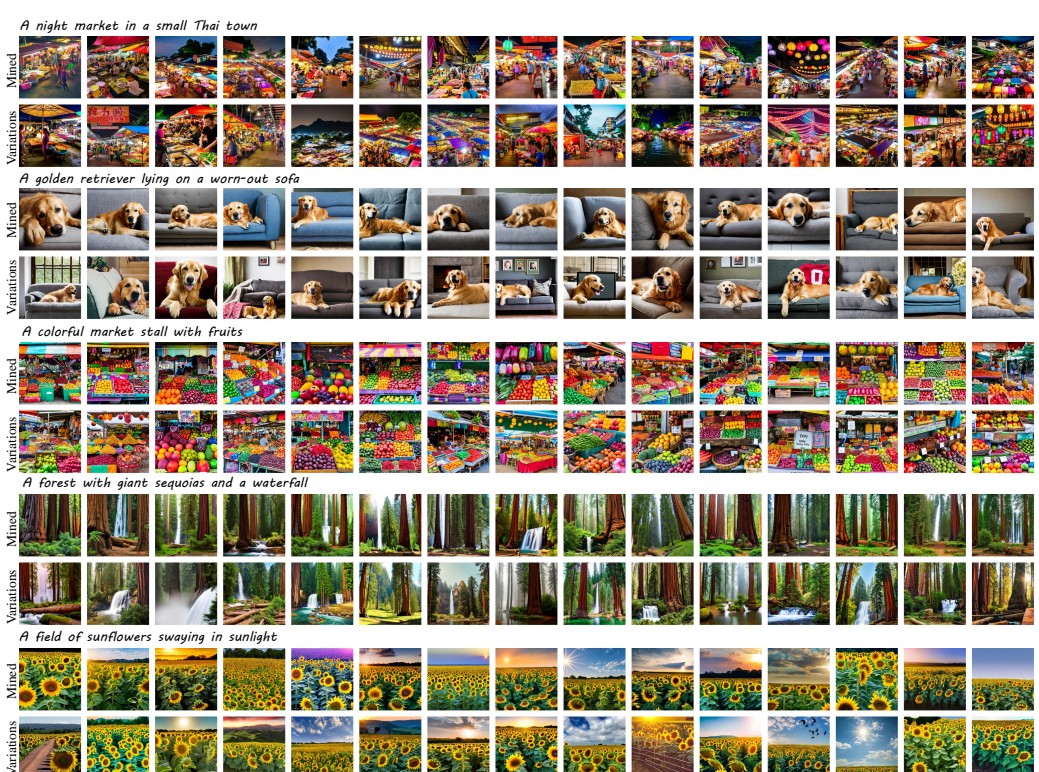

Figure S14: **Visualization of the missed visual concepts in prompts mined for SD 2.1.**

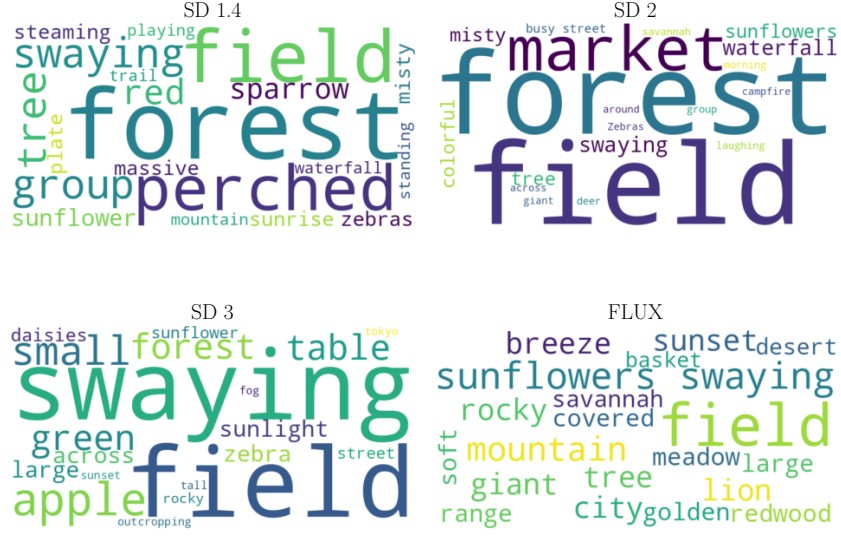

Figure S15: **Recurring terms in mined prompts.** Word clouds generated from the 50 most biased prompts mined for each model. The recurrence of words that are related to nature such as field, forest and green, suggests that these terms often result in limited visual diversity.

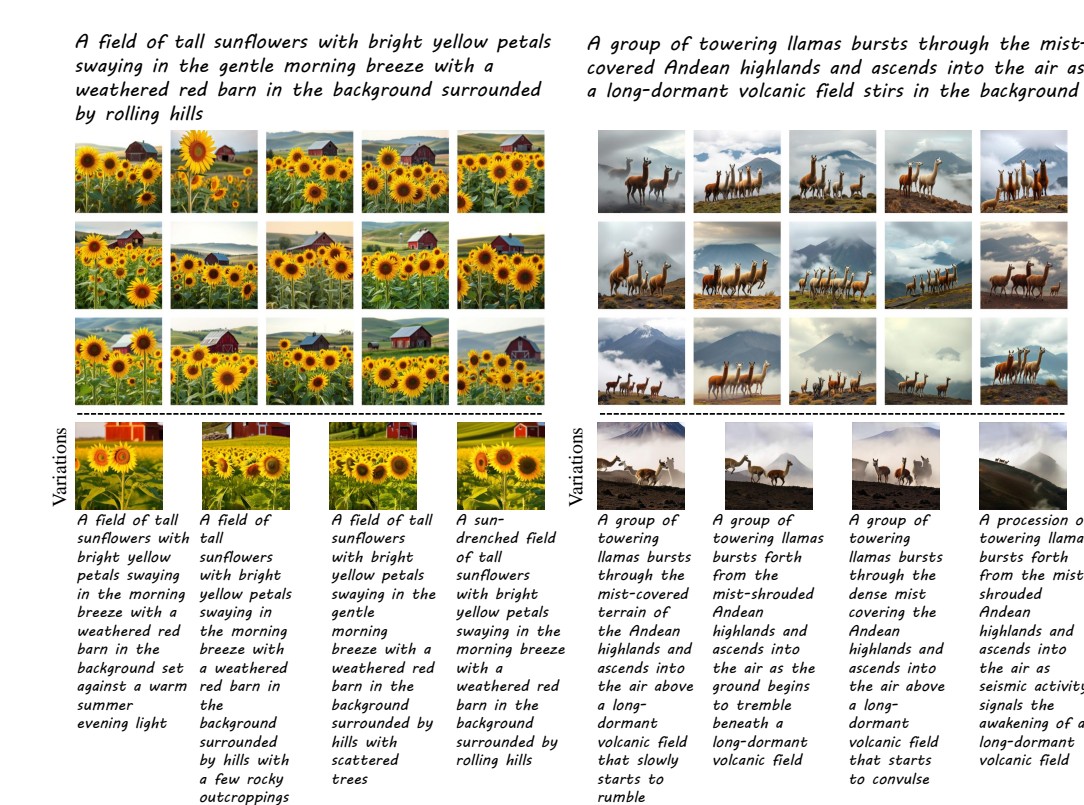

Figure S16: **Visualization of the missed visual concepts in prompts mined when relaxing the constraint on prompt length.** Prompts mined for FLUX.1 Schnell.

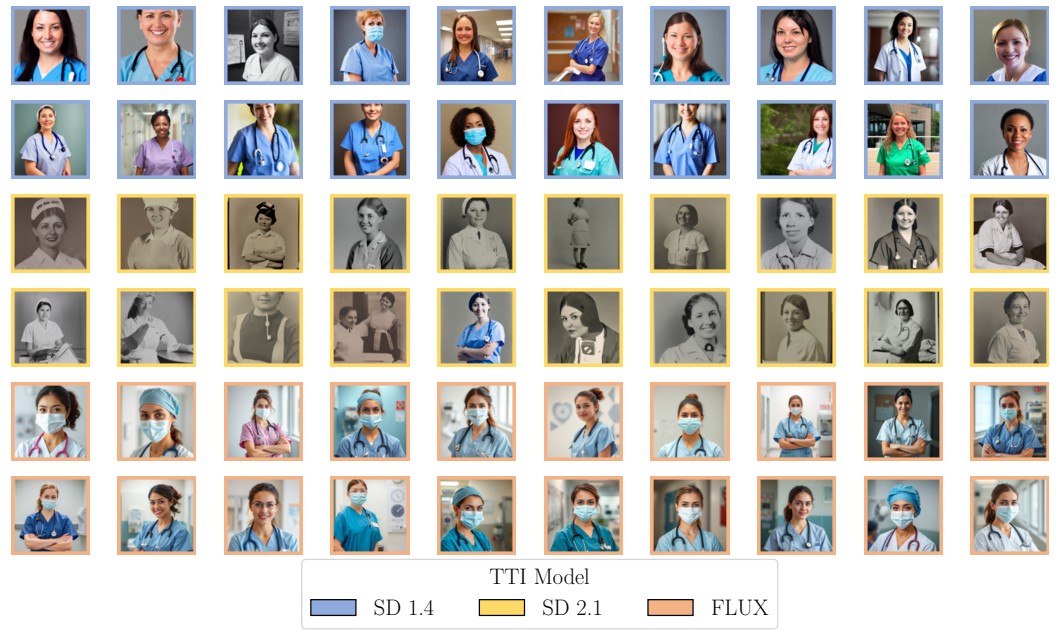

Figure S17: **Images generated with the prompt "a photo of a nurse".** Legend indicates TTI model used to generate the images.

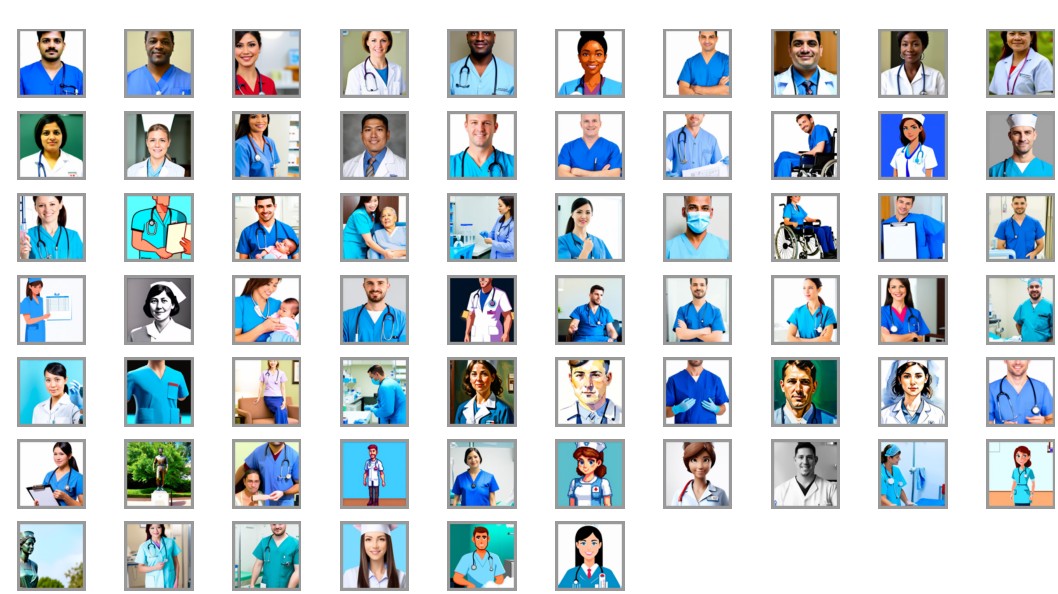

Figure S18: **Images generated with variations of the prompt "a photo of a nurse".** All images were generated with SD 3.

### D.1 MINING WITH DIFFERENT LLMS

Since MINETHEGAP is agnostic to the choice of LLM, we evaluate its behavior when driven by different models. We run 20 mining processes for FLUX.1 Schnell driven by LLaDA 8B-Instruct (Nie et al., 2025) and 20 driven by Qwen 2.5-7B-Instruct (Yang et al., 2024), and compare them with the runs driven by Llama 3.1–8B–Instruct. For each run, we use the $K = 5$ most biased prompts, yielding 100 prompts per LLM. As a baseline, we also sample 100 captions from COCO. We embed all four sets (Qwen-mined, Llama-mined, LLaDA-mined, COCO captions) using CLIP and compute the mean nearest-neighbor cosine similarity between every pair of sets, as reported in Fig. S19. We find that prompts mined by the three LLMs are more similar to each other than to random captions from COCO. To further analyze the distributions, in Fig. S20 we project the CLIP embeddings of all sets into 2D using t-SNE. The mined prompts from different LLMs exhibit relatively similar distributions, while COCO captions occupy sub-regions of the semantic space.

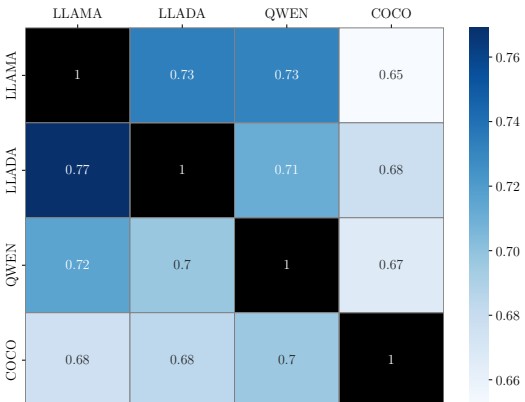

Figure S19: **Nearest-neighbor similarity of prompt sets.** Mean cosine similarity to nearest-neighbor between prompts mined with Llama, LLaDa and Qwen, and captions from COCO. Prompts mined with different LLMs are more similar to each other than to COCO captions.

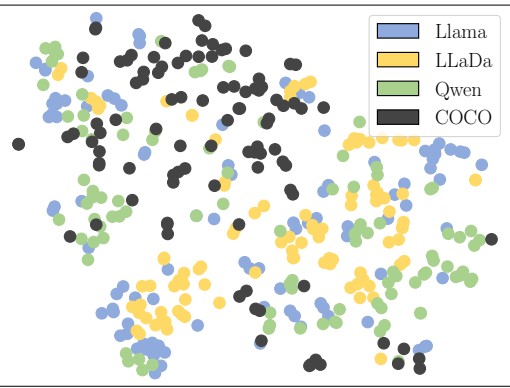

Figure S20: **t-SNE visualization of prompts.** 2D projection of CLIP embeddings for mined prompts (Llama, LLaDA, Qwen) and COCO captions. Mined prompts form overlapping distributions with some clustering, while COCO captions occupy subregions of the space.

### D.2 MINING SPECIFIC BIASES

To qualitatively demonstrate that MINETHEGAP can uncover particular types of bias, we adjusted the meta-prompts guiding the LLM used during the mining process. Specifically, for the discovery of sociodemograohic biases, while generating random prompts and prompt mutations we instructed the LLM (Llama 3.1–8B–Instruct) to depict people (*e.g.*, their roles, activities, professions, or social interactions). For generating the corresponding prompt variations, the meta-prompt instructs to retain the original meaning of the prompt but explicitly vary aspects such as gender, race, and age. This focused prompting allows our method to target specific sociodemographic biases. Figure S21 illustrates two examples: the left pane shows outputs generated by SD 3 for a mined prompt where all images depict male firefighters, while the right pane shows SD 1.4 generating images a female teacher. In contrast, the associated prompt variations generated by the LLM exhibit greater demographic diversity, including female firefighters and male teachers.

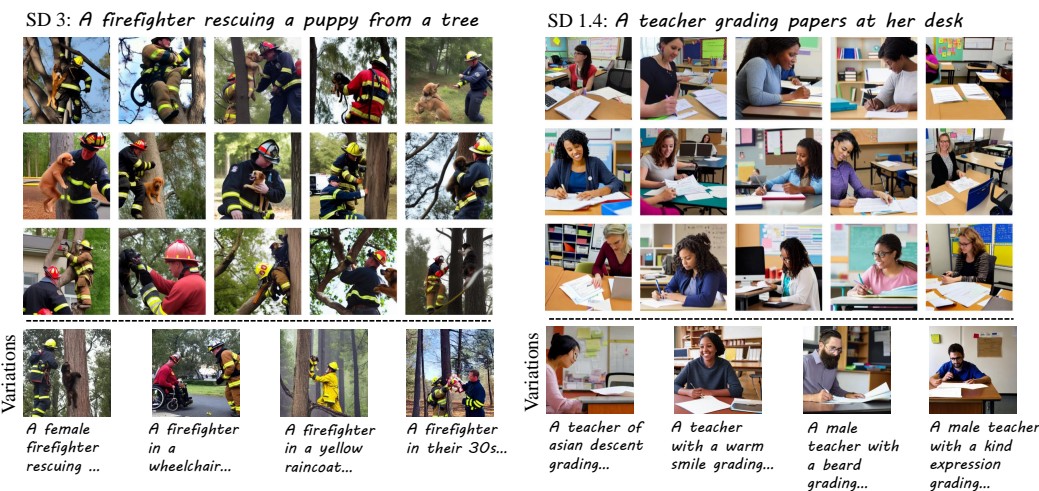

Figure S21: **Gender Bias.** By controlling the meta prompts used for Llama 3.1-8B-Instruct, we manage to capture specific biases such as gender and race. On the left pane, all images show a male firefighter although the prompts is gender-free. On the right pane, all images show a female teacher. The variations, on the other hand, are much more diverse including a female firefighter, a teacher of Asian decent, and a male teacher with a beard.

Figure S22 presents four examples of prompts illustrating specific target biases. To surface biases of specific object categories, the LLM was instructed to generate prompts that include an adequate object in both random and mutations phases, and instructed to vary features of the object and its setting during the text variations phase. For bias in the style of the outputs, only the meta-prompt instructing to generate variations differ from the open-set setting, asking to explore different image styles. Independent runs were conducted on general objects, food items, and clothing, and Fig. S22 mentions the setting each shown prompt was mined for. Thanks to the prompt variations which exhibit greater diversity under each subject, we manage to capture these biased prompts, highlighting the flexibility of our method in automatically mining specific kinds of representational biases when desired.

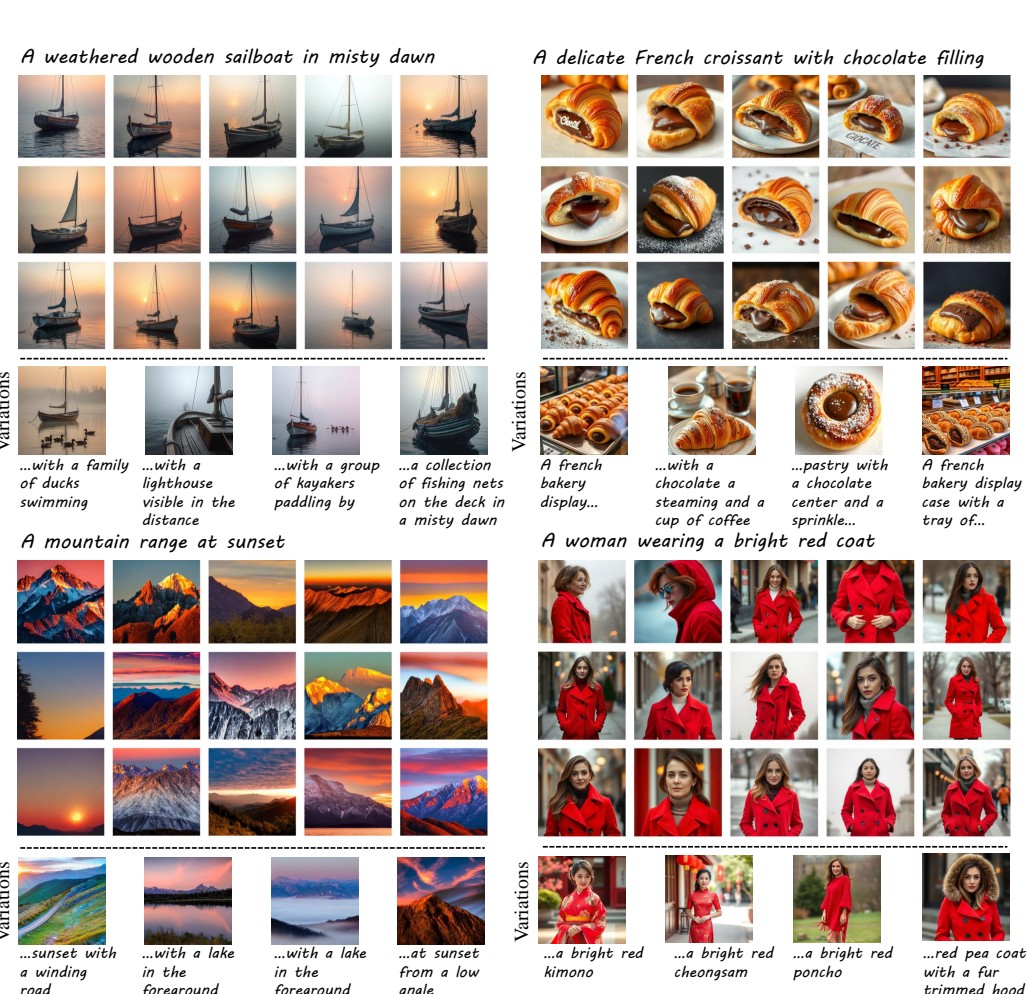

Figure S22: **Mining specific biases.** When aiming to reveal specific biases, controlling the meta prompts used for Llama 3.1-8B-Instruct enables capturing such targets. The figure shows four settings. Top row depicts mining prompts that exhibit bias when referring to objects (left) and food (right), while the bottom row show examples for image style (left) and clothing (right). The images generated to the variations which constitute the missed visual concepts reveal the source of the bias. For example, all the croissant images depict a single object from a similar angle, and all images of the woman exhibit the same style of coat.

### D.3 Supplementary results for the comparison with OpenBias

We provide additional qualitative comparisons with OpenBias. Figure S23 presents captions from COCO that were marked to exhibit high biases according to our bias score, while OpenBias failed to surface their biases. For example, in the prompt "A young boy with face paint all over his face," OpenBias proposed bias candidates related to the paint color as well as the child's gender and age. The VQA responses for the color candidates were nearly uniform, masking any detectable bias, and the age or race of the child could not be reliably inferred by the VQA. In contrast, our method generates diverse prompt variations such as specific signs or distinctive face paint patterns that expose the underlying bias more effectively, including:

- "A young boy with face paint and a big sign that says happy birthday"
- "A young boy with face paint and a big bow tie and suspenders"
- "A young boy with face paint resembling a tiger's stripes"
- "A young boy with face paint and a matching hat and cloak"

A similar pattern appears for the prompt "A person wears purple and black striped socks." OpenBias proposed only the sock color as a bias candidate and queried the VQA to decide whether the socks were purple or black. Our approach, however, produces more diverse variations that reveal richer bias cues, such as:

- "A person wears purple and black stripes socks while holding a coffee cup"
- "A person wearing purple and black stripes socks is looking at a phones"
- "A person wearing purple and black stripes socks is standing in front of a building"
- "A person wearing purple and black stripes socks is standing on a mountain trail"

are much more diverse and therefore enable capturing the bias of showing only the socks.

We further evaluate OpenBias on prompts mined using MINETHEGAP, to evaluate the extent to which it agrees on the presence of bias. Using our method, we extract 50 mined prompts from each model (*e.g.*, , SD 1.4, SD 2.1, SD 3, and FLUX) and provide them as input to OpenBias. Across these 200 textual prompts, OpenBias concludes the bias proposal stage with 19 unique bias names. Figure S24 illustrates several captions that OpenBias fails to identify as biased.

Consider the prompt "A hedgehog nestled in a handwoven rattan basket." OpenBias initially proposed the animal type and basket material as bias candidates but discarded them as integral parts of the caption. Our method, by contrast, generates diverse visual concepts that surface hidden biases, including:

- "A hedgehog snuggled in a handwoven rattan storage basket in a rustic cabin"
- "A hedgehog in a handwoven rattan planter basket surrounded by succulents"
- "A hedgehog nestled in a handwoven rattan picnic basket on a rocky outcropping"
- "A hedgehog in a handwoven rattan planter basket on a stone wall with ivy"

Finally, for the caption "A lioness walking in tall grass," OpenBias proposed bias candidates such as the animal itself, the grass height, and the color of the animal. These questions were difficult for the VQA to answer, and the visual similarity across images further obscured the bias. Our method again exposes richer variations, for example:

- "A lioness strolling in a landscape of tall grass"
- "A lioness walking in a grassy savannah of tall grass"
- "A lioness navigating through a thick stand of tall grass"
- "A lioness moving through a sea of tall grass"

*A person wears purple and black striped socks*

*A young boy with face paint all over his face*

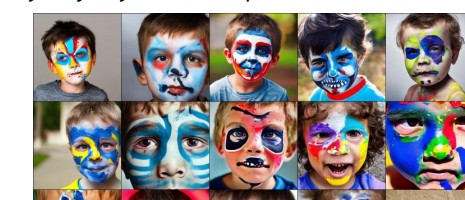

*There are many birds flying near the boat·*

*A close up of a man looking surprised·*

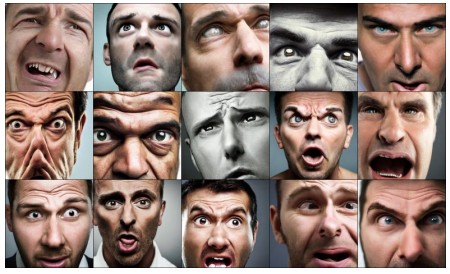

Figure S23: **Comparison with OpenBias on captions from COCO.** Visual examples of captions taken from the COCO dataset that are marked as biased by our bias score however overlooked by OpenBias.

*A lioness walking in a tall grass*

*A meadow of daisies swaying in the mist*

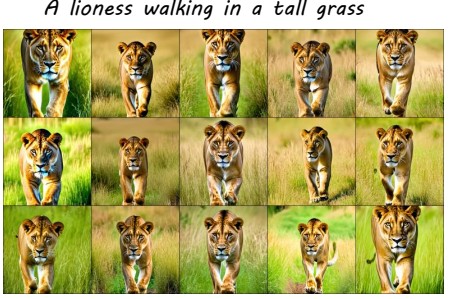

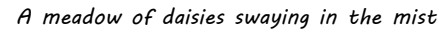

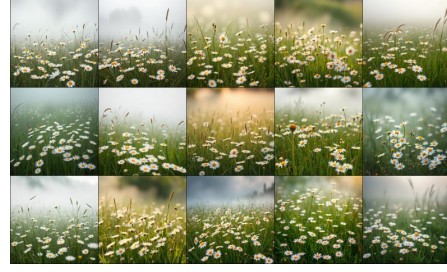

*A firefly perched on a bright orange marigold*

*A hedgehog nestled in a handwoven rattan basket*

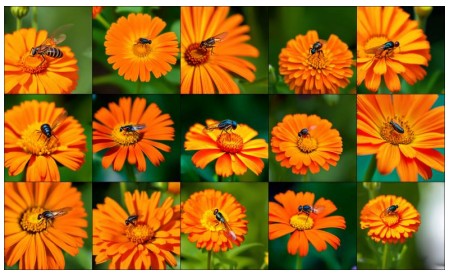

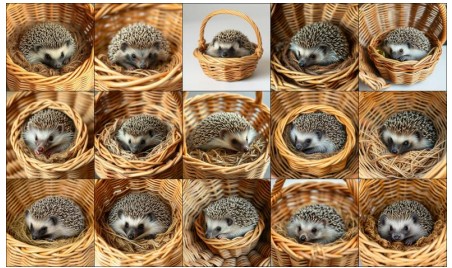

Figure S24: **Comparison with OpenBias on mined prompts.** Visual examples of prompts mined by MINETHEGAP however overlooked by OpenBias.

