# OpenReview forum: "MineTheGap: Automatic Mining of Biases in Text-to-Image Models"
_ICLR.cc/2026/Conference — Submitted to ICLR 2026_

### Official Review · Reviewer_MP9j · 2025-10-15

**Soundness:** 1
**Presentation:** 2
**Contribution:** 3
**Rating:** 4
**Confidence:** 3

**Summary:**

This paper introduces MineTheGap, a framework for automatically mining text prompts that cause TTI models to generate biased outputs. The method employs a genetic algorithm to iteratively search the prompt space, guided by a novel, open-set bias score. This score quantifies bias by comparing the distribution of images generated by the TTI model for a given prompt to a reference distribution of plausible interpretations. This reference distribution is approximated by generating a set of diverse "textual variations" of the original prompt using LLMs. A large gap between the generated image distribution and the textual variation distribution indicates a high degree of bias. The authors validate their bias score by demonstrating its correlation with real-world occupational statistics.

**Strengths:**

**1. Novel Bias Score for Open-Set Scenarios**

The paper introduces a novel and clever bias score that moves beyond traditional, axis-based evaluation. By quantifying bias as the distributional gap between generated images and LLM-generated textual variations, the method provides a flexible and powerful way to measure bias in open-set scenarios without needing predefined categories like gender or race. This is a significant methodological contribution to the field of fairness evaluation.

**2. An Automated Framework for Proactive Bias Mining**

It proposes a complete, automated framework called MineTheGap for mining problematic prompts from the vast text space. The use of a genetic algorithm to search for and optimize prompts that expose model weaknesses is a sound approach. This shifts the paradigm from passively detecting bias on a given set of prompts to actively discovering a model's inherent vulnerabilities.

**3. Strong Validation of the Proposed Metric**

A key strength of this work is the validation of its proposed bias score. The authors demonstrate that their metric's ranking of occupational bias shows a strong and positive correlation (Spearman's ρ = 0.72) with real-world labor statistics, even outperforming a notable prior method.

**Weaknesses:**

**1. Unproven Assumption of the Bias Score**

The paper's entire bias metric rests on a strong and unproven assumption: that an LLM's "textual variations" can serve as a valid proxy for the true, human-expected distribution of a prompt's meaning. This is a fundamental weakness, as the LLM used to generate this reference distribution is itself inherently biased and has a limited scope of creativity. While the authors briefly acknowledge this in the conclusion, the fact that the "ground truth" is actually another potentially flawed model significantly undermines the claimed objectivity of the bias score.

**2. High Computational Cost and Lack of Practicality Analysis**

The framework appears to be computationally expensive. The genetic algorithm requires generating N images and N text variations, followed by an N x N similarity calculation, for every candidate prompt in every iteration. The paper provides no discussion or estimate of the actual computational resources (e.g., GPU hours) for this process, making it difficult to assess the practical viability and scalability of the method as a real-world auditing tool.

**3. Lack of Sensitivity Analysis for the Bias Score's Hyperparameters**

The proposed bias score is dependent on key hyperparameters, notably the number of generated samples and the percentile choice. The paper does not provide any sensitivity analysis to demonstrate how the ranking of biased prompts would be affected by different choices of these hyperparameters. This is a significant omission that makes it difficult to assess the robustness and stability of the core metric.

**4. Insufficient Baselines for the Prompt Mining Framework**

While the paper validates its bias score against prior work, the genetic algorithm itself lacks a rigorous comparison against simpler baselines. For instance, it is unclear how much the complex evolutionary process outperforms a more straightforward approach, such as a large one-shot random sampling of prompts ranked by the same bias score. Without such a comparison, the added value of the genetic algorithm over simpler search strategies is not well-established.

**Questions:**

**1.** The validity of the proposed bias score hinges on the assumption that an LLM's output can serve as a proxy for the true human-expected distribution. Given that any LLM is itself inherently biased, could the authors provide a stronger justification for this core assumption or discuss how the choice of LLM might influence the discovered biases?

**2.** The paper lacks any analysis of the computational cost, which appears to be a significant concern for the method's practicality. Could the authors provide an experiment or at least an estimate of the resources (e.g., total GPU hours) required for a full mining run on a single model and comment on the framework's scalability for real-world auditing?

**3.** The stability of the bias score is dependent on key hyperparameters. Could the authors provide a sensitivity analysis experiment to demonstrate how the ranking of biased prompts is affected by variations in these crucial parameters, in order to validate the metric's robustness?

---

> ### Author Response · Authors · 2025-11-20
>
> We appreciate the points raised and have addressed them below.
>
> **Unproven Assumption of the Bias Score:**
>
> We acknowledge that the diversity generated by the LLM is essential for our bias score and that LLMs themselves may introduce biases. However, we emphasize that text variations are not sampled using the LLM as i.i.d samples, but as a response to a single meta-prompt which focuses on diversity. When targeting a specific bias with a known set of interpretations, alternative constructions (such as the Cartesian product in Sec. 4.2) can be used instead. In the open-set setting, however, there is no ground-truth distribution of interpretations, and it is therefore standard practice to rely on LLMs or VLMs to propose semantic variations[1].
> To further support this choice, we conducted a user study, presented in Sec. 5, showing strong alignment between human judgments of bias and the bias score assigned to mined prompts, indicating that LLM-generated variations provide a reasonable proxy in our framework.
>
> **High Computational Cost and Lack of Practicality Analysis:**
>
> Evaluating a large number of prompts is computationally demanding, however, this approach is still far more efficient and systematic than manual evaluation. Without optimizing for runtime, on average experiments take between 7 minutes per iteration on SD 2.1 to 12 minutes on FLUX.1 Schnell, resulting in 3 to 5 hours  when running 25 iterations, on a single NVIDIA RTX A600 GPU.
> We emphasize that comparing distributions enables generating a single set of images for a given prompt (as opposed to methods which generate multiple sets for each axis of bias), prompt an LLM once for textual variations (as opposed to asking for bias axes and for counterfactuals or questions for each axes) and compare them with CLIP (rather than querying a VLM multiple times). Nevertheless, to keep evaluations tractable, we use relatively lightweight models: CLIP for image-text comparisons and LLaMA 3.1–8B–Instruct as a reasonably modest LLM. We believe this setup offers a good balance between computational efficiency and semantic sensitivity. Furthermore, when the targeted bias is more specific, fewer iterations should be sufficient to uncover biases. For instance in the extreme case of a simple task of finding prompts that produce images of a specific color, the algorithm converged in just four iterations (see Appendix A).
>
> **Lack of Sensitivity Analysis for the Bias Score's Hyperparameters:**
>
> Following the reviewer’s concern, we have added an analysis of the effect of $\alpha$ to App. A. The essence of the hyper parameter $\alpha$ is to control the sensitivity of the overlap between the two distributions being compared. We show in Fig. S5 that different values for $\alpha$ have positive correlation with the human preferences collected in the user study, and the peak is at $\alpha=25%$, which is the configuration we used in our experiments.
>
> **Insufficient Baselines for the Prompt Mining Framework:**
>
> We appreciate the reviewer’s point. A comparison against the suggested simpler search strategy was included in the initial submission (App. A), where we evaluated the genetic optimization process against an equally sized random sample generated by an LLM, as well as against random samples from the COCO dataset, all scored using the same bias metric. The results showed a clear superiority of the mining procedure, demonstrating that it provides added value beyond straightforward random sampling.
> We additionally conducted a user study evaluating the entire mining framework, where mined prompts were consistently perceived as more biased, between 69% and 81% of responses favoring the random prompt in each model (Line 477).
>
> [1] Chinchure et al., Tibet: Identifying and evaluating biases in text-to-image generative models. ECCV 2024.

---

> ### Comment · Reviewer_MP9j · 2025-11-21
>
> Not all of my questions are resolved. I will wait the rersponse of other reviewers and may change my idea after the discussion with them. But now I choose to keep my overall assessment.

---

> > ### Author Response · Authors · 2025-11-23
> >
> > We appreciate the reviewer's response, and will be glad to know which questions were not resolved so that we can provide additional explanation.

---

### Official Review · Reviewer_Bze9 · 2025-10-20

**Soundness:** 3
**Presentation:** 3
**Contribution:** 3
**Rating:** 6
**Confidence:** 5

**Summary:**

The paper introduces a genetic-algorithm framework that mines prompts likely to expose bias in text-to-image models. It scores each prompt by comparing a batch of generated images to LLM-produced text variations in a shared embedding space, using a bias.-score that depends on coverage and relevance which is then used in the algorithm to identify prompts that induce biases. Experiments are performed across SD 1.4, SD 2.1, SD 3, and FLUX.1 models.

**Strengths:**

1. The approach goes beyond fixed biased and tackle an important and challenging problem  automatic mining of prompts that induce biases.
2. The experimental results are robust, spanning SD 1.4/2.1/3 and FLUX models, which suggests the approach is architecture-agnostic.

**Weaknesses:**

1. The bias score considered cannot be considered a parity metric. it doesn’t check how often each group appears, only whether every text considered has at least one matching image (coverage) and every image matches some text (relevance). I think there is a loophole here.  For example, with the texts “female doctor,” “female doctor in ER,” “male doctor,” and “male doctor in office,” a batch containing three male office portraits and one female ER photo will still “pass” both coverage and relevance where each row has a single good match, and each image has a matching text. The score would be high (suggesting low bias) despite an obviously skewed distribution (3/4 male). I believe that this metric may underreport demographic imbalance (or any other imbalances) compared to parity-style measures (e.g., those used in OpenBias) that directly assess group proportions.

2. This is again related to the first point. In Section 4.2,  the authors force demographic sensitivity by constructing text variations via a gender×race Cartesian product, which makes the score responsive to demographic omissions. But if the text variations are randomly generated without explicitly targeting gender or race, as in the “doctor” variations listed in Appendix C.3, gender bias may not surface as a “missed visual concept.” With only a small number of female-specific texts and some female images, the evaluation can still show good coverage and relevance, effectively missing a known demographic skew. Hence, this method has limitations in detecting certain known biases, especially representational imbalances, unless demographic attributes are explicitly and sufficiently encoded in the text set. In the “unknown-bias” setting, where no ground truth exists, this makes it hard to assess reliability and the reported results may not faithfully reflect real-world bias.

3. It would have been stronger if the paper had included a user study for the unknown-bias detection setting, to measure how well the method’s findings align with human judgments.

**Questions:**

1. What happens if you use a parity metric  as in OpenBias instead of the bias score introduced in the paper?
2. Could the authors report the missed visual concepts for the prompt “a doctor” using the textual variations from Appendix C.3? Given the known biases in this case, this will provide a validation the above points.
3. The method is dependent on multiple hyperparamaters. Would the same set of hyperparameters work for other cases where prompts are more complex?

---

> ### Author Response · Authors · 2025-11-20
>
> We thank the reviewer for the constructive review.
>
> **Parity metric and imbalances:**
>
> We appreciate the reviewer’s insightful observation. It is correct that our metric measures coverage and relevance rather than proportion, and therefore does not function as a parity-style metric. This is not different from OpenBias, as we explain below. This choice reflects the open-set setting we target: when a prompt contains multiple ambiguities across many potential bias axes, and the setting is generating a batch of images to be presented to the user, we set our goal to determine whether each concept is represented at least once. In this context, a single image per concept is sufficient for detecting the presence of an open-set bias. When targeting proportional deviations such as gender imbalance, the textual variations can be constructed to encode the desired proportions explicitly, as done in Sec. 4.2. By using a Cartesian combination of gender and race, our score was superior to applying OpenBias in the BLS setting.  Even for socio-demographic attributes, we allow a single matching image because our objective is to span multiple aspects of these attributes rather than quantify their relative frequencies.
> As suggested by the reviewer, and to clarify this point, we ran our bias score on photos generated for the prompt “a photo of a doctor” (see App. C3 and Fig S9). We generated a set of 15 images using four TTI models, and used the same set of variations mentioned in the text across all sets. All images generated by FLUX.1 Schnell depict male doctors, and the variation describing a female is indeed surfaced as a missed visual concept. However, image sets generated with the other models include at least one depiction of a female doctor, and although females appear in different proportions in all three sets, other semantic aspects from the textual variations that fail to appear in the images are surfaced instead.
> Note that in OpenBias, a model is considered unbiased with respect to a specific concept if “the possible classes exhibit a uniform distribution”. Restricting the evaluation to a uniform distribution fails to capture cases where the distribution should intentionally be skewed, as in evaluating the age of the figure generated for “a pregnant woman”.
>
>
> **User study for the unknown-bias detection setting:**
>
> To further strengthen our evaluation, as the reviewer proposed, we conducted a user study to measure the alignment of the method’s findings with human judgments, reported in the updated version (Line 477). We collected sets of random prompts and mined prompts for each model, and conducted a paired test where participants were shown two prompts, one from each group, along with a set of 15 images generated for each prompt. In each comparison users were instructed to choose the prompt whose image set appeared less biased with respect to attributes not specified in the prompt. The results show that mined prompts were consistently perceived as more biased, between 69% and 81% of responses in each model.
> In a further analysis of the results, we compared the score difference between the two prompts in each question with the fraction of users who favored the corresponding prompt, assuming that if the distance is larger there should be a larger agreement and vice versa. This was shown to be the case, with a Pearson correlation of 0.71.
>
>
> **Hyperparameters for more complex prompts:**
>
> More complex prompts naturally contain less ambiguity, yet they still leave certain aspects unspecified. We have added to App. D examples for more complex prompts mined when targeting prompts with up to 30 words. Our work intentionally focused on relatively short prompts to uncover general biases, however, the method is not limited to shorter ones as during the variation phase, the LLM is instructed to explore any remaining ambiguity, allowing it to adapt to whatever aspects remain underspecified. This enables the method to reveal biases even when they emerge under more specific prompts, as we show in the examples added. We do not expect the hyperparameters of the genetic optimization, as the balance between mutated and random candidates at each iteration, to require adjustment in such cases.

---

### Official Review · Reviewer_EUCN · 2025-11-01

**Soundness:** 3
**Presentation:** 2
**Contribution:** 2
**Rating:** 4
**Confidence:** 4

**Summary:**

* The paper introduces MineTheGap, an automatic prompt mining method to uncover instructions that would lead to bias in Text-to-Image (TTI) models' generations.
  * The method uses a genetic algorithm and a LLM to iteratively refine a pool of prompts, enabling the discovery of both known and previously unseen open-set biases.
* The author proposes a novel bias score that ranks biases according to their severity to assist with the optimization process.
  * The bias measurement method compares the distribution of generated images to distributions of textual variations to capture deviations.

**Strengths:**

* The automatic prompt mining technique is novel in TTI model bias evaluation.
* The bias measurement method by ranking prompts based on bias score is also novel and seems to be effective from the visual examples.

**Weaknesses:**

My concerns about this paper is mainly regarding its problem formulation, experiment design and empirical results. While the core methodology is novel, the overall experiment design and result analysis lacks systematic rigor. The current presentation makes it difficult to fully assess the efficacy and generalizability of the proposed approach across diverse scenarios and TTI architectures.

* **Ambiguity in Bias Definition and Scope**: The paper does not provide a clear, formal definition of "bias" within the context of Text-to-Image (TTI) models, nor does it explicitly delineate the categories or dimensions of bias the proposed framework is designed to detect. Clarifying the theoretical boundaries of the framework is essential.
* **Reliance on Un-Benchmarked Prompt Initialization**: The methodology's dependence on LLM-sampled initial prompts and a self-designed prompt refinement paradigm without a clearly stated problem definition (i.e., the specific types of bias targeted for exploration) introduces an element of subjectivity. A more robust approach would involve starting with prompts derived from existing, established bias benchmarks to provide a grounded point of comparison.
* **Questionable Scalability to Complex Prompts**: The reliance on overly simple, short prompts (limited to eight words) raises significant doubts about the method's effectiveness on more recent and powerful TTI models capable of processing longer, more complex, and semantically sophisticated instructions. The paper should include a thorough analysis of how the method performs with the scaling of prompt length and complexity.
* **Narrow Validation of Bias Score and Lack of Human Grounding**: The validation of the new bias score is primarily demonstrated on a constrained and simple task (gender bias in "portrait of profession" prompts) using only a Spearman rank correlation. It remains unclear how the score generalizes to other domains of bias or more complex prompts. Furthermore, the absence of a human validation study to verify the perceptual relevance of the bias score is a notable oversight.
* **Insufficient  Quantitative Results**: While the paper is proposing a new bias score metric, the authors provide limited quantitative results on different TTI models or bias dimensions. The current experimental section relies heavily on qualitative examples and visual demonstrations. Without showing the bias in TTI models that your method is successful in capturing, I don't think the main contribution can be justified.
* **Insufficient Model Evaluation**: The authors only conducted experiments on a limited set of models, omitting tests on several state-of-the-art TTI models, such as the publicly available Qwen-Image or leading closed-source models like DALL-E and GPT-Image-1. A comprehensive audit requires testing against the current frontier of TTI models.
* Additionally, the image examples are small and hard to see, especially in Figure 2.

**Questions:**

Please see Weakness.

---

> ### Author Response · Authors · 2025-11-20
>
> We thank the reviewer for the thorough review and feedback.
>
> **Ambiguity in Bias Definition and Scope:**
>
> This is a good point. The framework is intentionally designed to cover broad dimensions of bias, where the main essence is ranking the prompts by quantifying the discrepancy $d(P,Q)$ between two sets: $Q$, the set of $N$ images generated by a TTI model for the prompt, and the target set $P$, of $N$ possible interpretations of the same prompt.
> We approximate $P$ using textual variations generated by an LLM, which is prompted to explore ambiguities and span diverse semantic options. The correspondence between $P$ and $Q$ is then measured in a shared embedding space using CLIP.
>
> This formulation allows us to rank prompts by their overall bias, even when the bias arises across multiple categories, as in open-set settings where the LLM explores uncategorized semantic dimensions. It can also be adapted to target specific axes of bias, as done in the closed-set settings (App. D.2) , by instructing the LLM to generate variations constrained to these predefined interpretations (as socio-demographic ones).
>
> **Reliance on Un-Benchmarked Prompt Initialization:**
>
> Indeed, our method introduces LLM based randomness both in the initial prompt selection and in the randomly injected prompts during optimization. This design choice is intentional: our goal is to detect open-set biases without assuming any prior knowledge about their types or forms. Under this setting, random prompts provide the most neutral and assumption-free initialization for exploring the space of possible prompts. When targeting specific types of bias, our method still initializes from random prompts, yet the LLM is instructed to sample only ones that are relevant to the specific bias (e.g., for socio-demographic biases the prompt needs to include at least a person).
>
> An alternative initialization strategy could be to use COCO-based prompts (as done in OpenBias). However, as shown in App. A, Fig. S.2, random prompts actually exhibit stronger bias signals than COCO prompts, making them a more suitable starting point for open-ended bias discovery. Finally, we note that there is currently no established benchmark for open-set bias evaluation. Existing bias datasets focus on predefined, known biases, and we therefore evaluate on those to provide a point of comparison, even though they only partially reflect the open-set nature of our setting.
>
> **Questionable Scalability to Complex Prompts:**
>
> Prompt length plays an important role in the expected level of bias, as the more specific or detailed a prompt is, the less room there is for interpretation (and biases). We have added examples for applying MineTheGap for longer prompts to App. D (Fig. S16). While our framework is independent of the length of the prompt, to better surface general biases, we deliberately targeted relatively short prompts, which is in line with other works in the field which for example evaluate prompts from COCO[1], or a combine prompts written by the authors with simplified versions of prompts from Diffusion[2], all intentionally short.
> Nevertheless, since in the variation phase the LLM is instructed to explore any ambiguity remaining in the prompt when generating variations, it naturally adapts to aspects left unspecified. This allows the method to reveal biases also if they emerge even under more specific prompts.

---

> ### Author Response · Authors · 2025-11-20
>
> **Narrow Validation of Bias Score and Lack of Human Grounding:**
>
> Following the reviewer’s recommendation, we conducted a user study to provide an additional validation of the bias score, reported in Sec. 5. We collected sets of random prompts and mined prompts for each model, and conducted a paired test where participants were shown pairs of prompts, one from each group, along with corresponding sets of 15 images generated for each prompt. In each comparison users were instructed to choose the prompt whose image set appeared less biased with respect to attributes not specified in the prompt. Across the four evaluated TTI models, users consistently ranked mined prompts as more biased than random prompts.
> To validate the bias score, we compared the score difference between the two prompts in each question with the fraction of users who favored the corresponding prompt, assuming that if the distance is larger there should be a larger agreement and vice versa. This was shown to be the case, with a Pearson correlation of 0.71 (Fig. 11).
>
> In addition, we adopted two complementary evaluation strategies. First, to assess bias ranking, in Sec. 4.2. we compared the ordering induced by our score on known biases to ground-truth occupational statistics, showing improved alignment relative to the baseline. Second, to evaluate behavior in the open-set setting, we used our score to quantify diversity within generated image sets, examining how our score responds to (1) changes in CFG, known to modulate diversity (Sec. 4.2), and (2) generating images with and without CADS[3], a method established to enhance diversity in diffusion-based TTI models (App. B.2). Both showed the expected trend.
>
> **Insufficient Quantitative Results:**
>
> The user study described above was used to evaluate the entire mining framework. The mined prompts were consistently perceived as more biased than random prompts, between 69% and 81% of responses choosing the mined prompt as more biased than the random prompts (Fig. 10).
> When analyzing questions individually, only four out of 40 questions had a majority vote for the mined prompt being less biased. This supports the finding that the mining procedure successfully identifies prompts that induce biased generations in an open-set setting.
> We refer to the missed visual concepts as a source of explanation for the discovered bias, and will add a summary of the discovered biases to the final version.
>
> **Insufficient Model Evaluation:**
>
> We thank the reviewer for pointing out Qwen-Image, which was released not long before the submission deadline and has received significant attention for its state-of-the-art performance. We have now added qualitative results for this model alongside the four models originally included in our experiments in Fig. S7, showing prompts mined for Qwen-Image which exhibit strong biases. We will include additional comparisons in the final version. While our framework is compatible with any black-box TTI model, our focus in this work is on open-source models to ensure reproducibility and transparency.
>
> **Image examples:**
>
> We thank the reviewer for noting this. We have added  Figures S17 and S18 to the appendix, showing all images from Fig. 2, color coded as in the original plot. We used vector-graphics PDF in all figures to allow zooming and inspection of finer details.
>
> [1] D'Incà et al., Openbias: Open-set bias detection in text-to-image generative models. CVPR 2024.
>
> [2] Chinchure et al., Tibet: Identifying and evaluating biases in text-to-image generative models. ECCV 2024.‏
>
> [3] Sadat et al., CADS: Unleashing the Diversity of Diffusion Models through Condition-Annealed Sampling. ICLR 2024.

---

### Official Review · Reviewer_iZmd · 2025-11-01

**Soundness:** 2
**Presentation:** 3
**Contribution:** 2
**Rating:** 2
**Confidence:** 5

**Summary:**

This paper introduces MineTheGap, a framework for automatically discovering biased prompts in text-to-image (TTI) models. The method employs a genetic algorithm guided by a bias score that measures the extent to which a model’s generated image distribution deviates from the expected semantic diversity of a given text prompt. This expected diversity is approximated through LLM-generated textual variations.

**Strengths:**

- The paper presents an interesting idea that moves beyond traditional bias detection toward bias discovery, introducing an automated framework for mining prompts that reveal hidden or emergent biases in text-to-image models.
- Demonstrates the framework across multiple TTI models (Stable Diffusion 1.4–3 and FLUX), validating the metric through correlation with real-world statistics and showing model-specific bias discovery.

**Weaknesses:**

- The method relies heavily on LLMs to generate prompts and mutations, making its performance dependent on the quality and biases of the chosen LLM. Although the authors acknowledge this limitation, they do not analyze how different LLMs might affect the results or the stability of the mining process.
- The proposed metric depends on CLIP embeddings, which can carry demographic biases of their own. Because it measures similarity in this embedding space, it may wrongly treat harmless stylistic or compositional differences as bias. For example, two images of doctors in different lighting or art styles might be judged as “biased” simply because their embeddings differ.
- The choices of hyperparameters are not conceptually motivated or empirically ablated, such as the 20% percentile (α), number of iterations, and population size. Although ablations are included, they do not address parameter sensitivity or convergence behavior. Similarly, the construction of the initial population is unclear: it depends on LLM-generated prompts, yet the paper does not examine how variations in this initial set influence the outcomes. The role of random sentence injection is also insufficiently studied, leaving its contribution to exploration unexplained. Moreover, the mutation diversity relies entirely on the LLM’s ability to generate varied prompts. The paper does not verify whether these mutations are genuinely diverse or merely superficial rephrasings. For example, if the prompt “a cat on a chair” repeatedly mutates into “a kitten on a chair” or “a cat sitting on a chair,” the process would appear active but contribute little semantic diversity, potentially skewing the biased search results.

**Questions:**

- L11: "Text-to-Image (TTI) models generate images based on text prompts, which often leave certain aspects of the desired image ambiguous." What are these aspects?
- L51: "Prior approaches are limited to quantifying bias along specific axes and dividing each axis into a finite set of options, regardless if these are predefined as is typical in studies of sociodemographic biases, or proposed on the fly using LLMs." How is generating biases on the fly restrictive, especially in [1]? The authors argue that generating biases on the fly, as in [1], is restrictive because it limits the exploration to predefined bias axes. However, this claim appears inconsistent with their own approach as they also rely on predefined prompt templates containing specific bias structures (Appendix C.1). This reliance effectively mirrors the same constraint they critique, making their argument somewhat counterintuitive.
- L121: "These images (shown in (c)) were generated using SD 3 with textual variations of the prompt that specify gender, race, style, and surroundings." How is this process implemented? It appears to involve manual intervention, which raises concerns about scalability and automation.
- L159: "The optimization begins with an initial population of b diverse prompts, designed to broadly cover the space of possible biases. Ideally, this population approximates a uniform sample from the space of valid prompts, P." What is the method’s dependency on the initial prompt set? This should be quantitatively analyzed. Also, when the authors mention a “uniform sample,” do they mean sampling from a uniform probability distribution over the prompt space ( P )?
- L175: "We then generate mutations for each of the top-ranked sentences and add random sentences to form the next population." The effect of adding random sentences should be systematically analyzed, including the degree of randomness introduced and its impact on the search process. It would also be useful to examine whether these random samples act as soft or hard negatives, and how each type influences optimization behavior.

[1] Aditya Chinchure, Pushkar Shukla, Gaurav Bhatt, Kiri Salij, Kartik Hosanagar, Leonid Sigal, and Matthew Turk. Tibet: Identifying and evaluating biases in text-to-image generative models. In European Conference on Computer Vision, pp. 429–446. Springer, 2025.

---

> ### Author Response · Authors · 2025-11-20
>
> We thank the reviewer for the insightful comments and questions.
>
>
> **Reliance on LLMs:**
>
> We acknowledge that the method’s performance depends on the underlying LLM, as its interpretation of prompt ambiguities drives the discovery process. However, as we showed in in App. D.1, the influence this has on the mining process is limited. Specifically, this appendix compares mining results obtained with Llama 3.1-8B-Instruct, LLaDA 8B-Instruct, and Qwen 2.5-7B-Instruct, alongside a baseline of random COCO captions. The analysis shows that sentences mined by all three models are more consistent with each other than with random captions from COCO, indicating robustness across LLM choices. It should also be noted that it is quite standard in the field of open-set biases to rely on LLMs, VQAs or VLMs when assessing presence of bias[1,2,3], e.g., for suggesting biases or creating counterfactual examples, although each of these models may potentially have its own biases. In our setting, while LLMs are not bias-free, their capacity for creative and diverse generation, when appropriately prompted, helps mitigate these concerns by effectively expanding underdefined aspects of a prompt. In particular, the textual variations are generated jointly in response to a single meta-prompt. For example, for the prompt “a dog” the variations span different breeds, contexts, and locations, without repeating the same semantic option, as might happen with i.i.d sampling. This produces a broader set of plausible interpretations.
>
>
> **Reliance on CLIP:**
>
> Comparing distributions of images and texts requires using a mutual embedding space, and we agree that the method to get these embeddings may carry biases of its own. To address this, we took two precautions. First, we evaluated correlations with a known bias ranking across three models and selected CLIP as the one showing the strongest alignment (Lines 327-333). Second, we aim to mitigate such biases by explicitly comparing against textual variations (e.g., instead of using CLIP to evaluate how similar an image is to “a chef,” we evaluate how similar it is to “an Asian chef”, “a female chef”, and so on). This approach pushes CLIP to differentiate between concrete alternatives, helping reduce the influence of its own biases.
>
> When exploring open-set biases or when mining specifically for artistic domains, a bias toward particular art styles can itself be an interesting finding. Given that CLIP is known to cluster images primarily by semantic content, we observe that explicitly mentioning the art style is necessary to properly surface and analyze stylistic biases, as done in App. D.2.
>
> **Hyperparameters and prompt population:**
>
> We agree with the reviewer that it is important to study the effects of the method’s hyperparameters. Kindly note that App. A analyzes many of these choices, but we also added further motivations for our choices following the reviewer’s comment (see Sec. 4.1), and additional ablations of population size and of $\alpha$ (App A.1). Fig. S5 shows correlation between the bias score when using different values for $\alpha$, and human judgments of bias collected in a user study (Line 477). Multiple values of $\alpha$ correlate positively with the human preferences, and the peak is at $\alpha=25%$, which is the configuration we used in our experiments.
>
> The ablation which was reported in App. A.1 assesses the parameters of the genetic setting, including the effect of injecting random candidates, from not injecting at all (random set to 0) to setting approximately half the population at each iteration to be random (7/15). Across models, five evaluated configurations performed similarly, and for consistency we used the setting that selects one-third of the prompts, generates two mutations each, and fills the remaining third with random candidates, balancing exploitation and exploration.
>
> For mutation diversity, the meta prompt used to instruct the LLM to generate mutations mentions explicitly “Avoid simple synonym substitutions or minor rephrasings that do not lead to a noticeable visual difference (e.g., changing “kid” to “child”).“ (Line 1119, App. C.1). We empirically find this instruction simple enough for the LLM to follow.

---

> ### Author Response · Authors · 2025-11-20
>
> Regarding the questions:
>
>
> * L11: The ambiguous aspects in TTI prompts can be diverse, ranging from socio-demographic attributes of people (age, gender, race), types or colors of objects and animals (specie of the butterfly, color of the hat), to elements of the scene (location, surroundings), and even stylistic features of the generated image (artistic style, point of view). Our method is designed to explore these underdefined aspects systematically.
>
> * L51: The intent in this line was that prior approaches are limited to quantifying bias along axes, meaning they measure each bias dimension independently. In contrast, our setting requires a way to rank prompts by their overall bias across all axes simultaneously. For example, while [1] provides extensive per-axis evaluations and a Top-K concepts analysis that considers multiple axes jointly, the latter is qualitative rather than quantitative, and thus cannot serve as the scoring function needed to guide our mining process. We have revised the text to clarify this distinction. As an additional clarification, the meta-prompts used to generate textual variations in Appendix C.1 (L1127) are intentionally vague regarding the structure of potential biases. They state: “The variations should retain the original meaning but explore different interpretations of any ambiguity in the original prompt. Variations could address any unspecified aspects of the subjects and of the style or setting of the image”. As a result, the variations span multiple ambiguities, while a variation can contain more than one axis.
>
> * L121: Figure 2 is not a method figure. It is only meant to provide intuition for how the distribution of images for a specific prompt varies across models. We do not advocate for running this process as a practical method. The generation of textual variations in this figure was more similar to that in Sec. 4.2 (where we validate the score) than to our mining method, which automatically generates text variations without intervention. Specifically, in this figure the texts were obtained by taking a Cartesian combination of race, gender, locations and surrounding options generated with an LLM, and style options taken from a prompt engineering template.
>
> * L159+L175: The initial prompt population together with the injected random candidates define the general semantic areas that are covered, while the distance evaluated from a random prompt depends on the number of mutations a random candidate “survived” (number of times its descendants were selected). Each run in our paper includes 120 random prompts (5 prompts in 24 iterations) on top of the initial population of 15 random prompts. Due to the large prompt space, each random initialization is likely to explore different regions and yield distinct biased prompts. This is visualized by the box plots in Fig. 9 showing the distributions of losses for mined prompts under different random seeds (the lowest boxplot for each model). The IQR and whisker range demonstrate the degree of variance introduced by randomness in the initialization and along the evaluation steps.
> Regarding the uniform sample in L159, it indeed refers to a uniform sample over the prompt space.
> Finally, we emphasize that in the setting of our genetic algorithm, the random candidate prompts are not negative examples. Each one of these prompts can potentially initiate a mutation trajectory that may yield a highly biased prompt, and some may even appear among the top-5 results. We collected this statistics, and random candidates appear in the top-5 results 16% of the time, while they comprise 36% of the overall populations evaluated. This indicates that while they can score highly on their own, their primary value lies in seeding mutation paths that evolve into high-bias prompts.
>
> [1] Chinchure et al., Tibet: Identifying and evaluating biases in text-to-image generative models. ECCV 2024.‏
>
> [2] D'Incà et al., Openbias: Open-set bias detection in text-to-image generative models. CVPR 2024.
>
> [3] Kim ei al., Discovering and mitigating visual biases through keyword explanation. CVPR 2024.

---

### Author Response · Authors · 2025-12-02

We thank the reviewers for the detailed reviews. We are glad the reviewers found our work innovative by **moving from bias detection to bias discovery** (iZmd, Bze9, MP9j).
Reviewers found both the proposed bias score and the mining technique to be **novel** (EUCN), and **highlighted the validation** of our method as a strength (iZmd, MP9j), including the effectiveness of the visual examples (EUCN). The experimental results were referred to as **robust**, evaluating four TTI models (iZmd, Bze9).

We improved the manuscript following the reviews, and highlight the main changes:

* **Human evaluation (L469, Figs. 10&11):** We conducted a user study to measure the alignment of the mined prompts, as well as the underlying bias score, with human judgement. Across the models, mined prompts were consistently perceived as more biased than the random ones, and the score differences were shown to correlate with human perception with a Pearson correlation of 0.71.
* **Ablation on score hyperparameter (L814, Fig. S5):** We added an analysis of the effect of $\\alpha$, which controls the sensitivity of the overlap between the distribution of generated images and that of the textual variations. Multiple values of $\\alpha$ correlate positively with the human preferences, and the peak is at $\\alpha=25%$, which is the configuration we used in our experiments.
* **Effect of population size (L805, Fig. S4):** We evaluated additional settings with a larger population size while keeping the same proportions of mutated and random prompts. Larger populations evaluate more candidates per iteration and therefore identify more biased prompts earlier in the optimization, at the cost of increased NFEs.
* **Scalability to longer prompts (L1284, Fig. S16):** We report examples for prompts discovered by MineTheGap when relaxing the constraint on prompt length, showing its scalability.
* **Mined categories (L481, Fig. 12):** We added an analysis on the bias categories for mined prompts, by evaluating the terms added in the missed visual concepts. Setting and nature related categories were found to be most frequent across all models.
* **Qwen-Image (Fig. S10):** Following the suggestion, we added mining results for Qwen-Image.

Although the reviewers did not participate in the discussion prior to November 26, we believe that we have addressed all concerns raised, adding new results and revising existing ones for clarity. We appreciate the AC’s efforts following the leakage, and are hopeful that our rebuttals will be taken into consideration.

---

### Meta-Review · Area_Chair_5sPR · 2026-01-08

**Summary:**

This paper introduces MineTheGap, a method for automatically discovering biased prompts in text-to-image models using a genetic algorithm guided by a bias score. Reviewers appreciated the shift from bias detection to bias discovery and found the idea interesting. However, major concerns were raised about methodological soundness, including heavy reliance on LLMs and CLIP embeddings (both potentially biased), insufficient theoretical grounding of the bias score, unclear hyperparameter sensitivity, and dependence on prompt initialization. One reviewer was strongly negative and confident in rejection.

**Reviewer Concerns:**

The authors added ablations, human evaluation, and robustness checks across LLMs, which improved clarity. Despite this, fundamental concerns persist: the bias metric may conflate stylistic variation with societal bias, the genetic search behavior is not well-characterized theoretically, and reliance on biased embedding spaces undermines conclusions. Disagreement between reviewers remained, with at least one firm reject unchanged after rebuttal.

**Reviewer Scores:**

Reviewer iZmd (2): Remains 2 (strong reject).

Other reviewers (4): Likely remain below threshold despite improvements.

---

### Decision · Program_Chairs · 2026-01-26

Reject